# Beyond Circuit Connections: A Non-Message Passing Graph Transformer Approach Towards Quantum Error Mitigation

**Tianyi Bao**[12†]**, Xinyu Ye**[1†]**, Hang Ruan**[1]**, Chang Liu**[1]**, Wenjie Wu**[1]**, Junchi Yan**[12*]

1. Sch. of Computer Science & Sch. of Artificial Intelligence, Shanghai Jiao Tong University
2. Shanghai Innovation Institute

`{btyll05,xinyu_ye,zzrh01,only-changer,wenjiewu,yanjunchi}@sjtu.edu.cn`

## Abstract

Despite the progress in quantum computing, one major bottleneck against the practical utility is its susceptibility to noise, which frequently occurs in current quantum systems. Existing quantum error mitigation (QEM) methods either lack generality to noise and circuit types or fail to capture the global dependencies of entire systems in addition to circuit structure. In this work, we first propose a unique circuit-to-graph encoding scheme with qubit-wise noisy measurement aggregated. Then, we introduce GTranQEM, a non-message passing graph transformer designed to mitigate errors in expected circuit measurement outcomes effectively. GTranQEM are equipped with a quantum-specific positional encoding, a structure matrix as attention bias guiding nonlocal aggregation, and a virtual quantum-representative node to further grasp graph representations, which guarantees to model the long-range entanglement. Experimental evaluations demonstrate that GTranQEM outperforms state-of-the-art QEM methods on both random and structured quantum circuits across noise types and scales among diverse settings.

## 1 Introduction

Quantum computing promises to transform fields like cryptography (Gisin et al., 2002) and machine learning (Biamonte et al., 2017) by solving problems intractable for classical computers. However, during the *Noisy Intermediate-Scale Quantum (NISQ)* era (Brooks, 2019), noises significantly impede its practical realization, limiting the performance and reliability of current quantum systems. While *Quantum Error Correction (QEC)* (Calderbank & Shor, 1996; Gottesman, 1997; Terhal, 2015) offers a theoretical framework for correcting errors through syndrome extraction and adaptive state updates, it usually requires a substantial number of additional qubits. This qubit overhead makes large-scale QEC implementations unfeasible with existing experimental hardware (Cai et al., 2023). Accordingly, *Quantum Error Mitigation (QEM)* has emerged as an alternative. Unlike QEC, QEM aims to produce near-noise-free results by algorithmically reducing noise-induced biases through post-processing, most without the need for extra qubits (Kandala et al., 2019; Bravyi et al., 2022; Daley et al., 2022; Kim et al., 2023; Liao et al., 2024). These strategies are crucial steps toward achieving near-term quantum utility and surpassing the capabilities of classical supercomputers.

Despite their promise, traditional non-learning-based QEM methods face practical limitations. For instance, Zero-Noise Extrapolation (ZNE) requires multiple executions of the same circuit at different noise levels to estimate the zero-noise limit, leading to exponential sampling overheads (Cai et al., 2023). Clifford Data Regression (CDR) (Czarnik et al., 2021) usually imposes restrictions on circuit structures by limiting the number of non-Clifford gates, and Probabilistic Error Cancellation (PEC) (Van den Berg et al., 2023) can be susceptible to certain types of noise in the system.

Recent advances in machine learning–based QEM methods have shown improved efficiency (Liao et al., 2024) and insensitivity to both circuit structure and noise types (Kim et al., 2020). However, most of them fail to effectively encode the structural information of quantum circuits, such as the

---

*Correspondence author, † Equal contribution. Work was in part supported by NSFC 62222607.

mitigators based on multilayer perceptrons (MLPs) (Kim et al., 2020) or ordinary least squares (OLS) (Liao et al., 2024). While those based on graph neural networks (GNNs) can model the topology of quantum circuits by representing them as graphs, they generally focus solely on *local* interactions through message-passing mechanisms that update node representations based on local neighborhoods but fail to capture the intrinsic nonlocality of *global* quantum systems (Bell, 1964; Brunner et al., 2014). Furthermore, a common drawback of these ML-based models is their strict reliance on ideal expectation values of training circuits as labels within the current paradigm, which can be challenging to obtain without classical simulations (Cai et al., 2023).

To address these challenges, we propose a novel machine learning-based QEM method beyond the circuit connections, capable of generalizing across diverse noise types and circuit structures. Unlike the approach of Kim et al. (2020), we develop a circuit-to-graph encoding scheme that effectively captures the structural information of quantum circuits. Compared to Liao et al. (2024), we introduce a non-message-passing graph transformer to handle global context within quantum systems in addition to circuit structure, circumventing the limitations of traditional GNNs. Furthermore, inspired by the mirror techniques for customized benchmarking (Proctor et al., 2022), we design a data augmentation scheme termed as circuit inverse composition, which reduces dependence on ideal expectation values of training data and refines the learning framework. Overall, our main contributions are as follows:

1) **Graph Encoding of Quantum Circuits:** We introduce a novel circuit-to-graph encoding scheme that effectively captures the structural information of quantum circuits. Circuit operations are represented as nodes with appropriately dimensioned features, qubit-wise noisy execution outcomes are embedded into edge features, and the overall system structure is encapsulated using a proposed quantum positional encoding.

2) **Quantum-Native Graph Transformer:** We tailor a graph transformer for QEM that captures both the circuit structure and intrinsic quantum nonlocality without relying on message passing. By leveraging multi-head attention mechanisms guided by a structure matrix encoding the graph topology, it effectively captures long-range correlations within the quantum system. We also introduce a learnable Quantum Circuit-Representative (QCR) node that adaptively aggregates global graph information. After training, the QCR node's representation serves as the input to the mitigator.

3) **Refinement of the Learning Framework:** We devise a data augmentation technique that constructs training data by composing circuits with their inverse circuits with random initial state assigned. It reduces the dependency on ideal expectation values in the training data, thereby refining the framework for learning-based models and extending their applicability for larger quantum systems.

4) **Extensive Experimental Validation:** We perform comprehensive experiments to assess the effectiveness of our proposed method. Our model demonstrates superior performance over both classical and machine learning-based approaches on random and Trotterized circuits. Additionally, we demonstrate that GTranQEM preserves effectiveness while training without circuits' ideal expectation values, underscoring its practical applicability.

## 2 PRELIMINARIES

### 2.1 BACKGROUND ON QUANTUM ERROR MITIGATION

Here, we briefly describe the key background related to QEM. Readers are referred to the seminal textbook (Nielsen & Chuang, 2002) for comprehensive details on quantum computing.

**Quantum State Vector.** It is represented by a **state vector** (i.e., **ket**) in a complex Hilbert space, encapsulating all the information about a quantum system. For a **single qubit**, residing in a 2-dimensional Hilbert space, the state vector is expressed as follows, with normalization $|\alpha|^2 + |\beta|^2 = 1$:

$$|\psi\rangle = \alpha|0\rangle + \beta|1\rangle,$$

where $|0\rangle$ and $|1\rangle$ are the computational basis states, and $\alpha, \beta \in \mathbb{C}$. In an $n$-qubit system, the state vector $|\psi\rangle \in \mathbb{C}^{2^n}$ is defined in a $2^n$-dimensional Hilbert space and can be written as:

$$|\psi\rangle = \alpha_1 |0 \cdots 0\rangle + \alpha_2 |0 \cdots 1\rangle + \cdots + \alpha_{2^n} |1 \cdots 1\rangle,$$

where the normalization condition is given by $\sum_{i=1}^{2^n} |\alpha_i|^2 = 1$, and $|0 \cdots 0\rangle = |0\rangle^{\otimes n}$ represents a **basis state**. Quantum states are manipulated using **quantum circuits (QC)**, which are constructed by applying a sequence of **quantum gates**.

**Expectation Value in Quantum Circuits.** Consider a quantum circuit represented by the unitary operation $U$ acting on an initial quantum state $|\psi_0\rangle$. To evaluate the performance and outcomes of a quantum circuit, it is essential to compute the **expectation value** of an observable $O$. The **ideal expectation value** of $O$ is mathematically defined as:

$$\langle O \rangle^{\text{ideal}} = \langle \psi_0 | U^\dagger O U | \psi_0 \rangle,$$

representing the expected measurement outcome of $O$ when the system evolves under the unitary operation $U$ without any noise. However, in practical scenarios, quantum systems are susceptible to various noise and errors due to decoherence, imperfect gate operations, and environmental disturbances. Consequently, the obtained expectation value, denoted as $\langle O \rangle^{\text{noisy}}$, deviates from the ideal value with a noise-induced bias (Cai et al., 2023). The primary objective of QEM is to reconstruct $\langle O \rangle^{\text{ideal}}$ from the noisy measurements $\langle O \rangle^{\text{noisy}}$, thereby enhancing the accuracy of quantum computations without the overhead of full quantum error correction.

## 2.2 QUANTUM DEVICE NOISE

Depending on whether the environment of the quantum system provides memory that retains correlations across system-environment interactions, the noise can be classified into *Markovian* and *non-Markovian* errors (Zhang et al., 2024).

A process that changes a system's state $\rho \to \rho'$ is considered Markovian if $\rho'$ depends only on $\rho$. If the error associated with a particular gate $g$ is *Markovian*, then it is described by some map $G_g : \rho \to \rho'$ that does not depend on the time, previous operations, or any other context variable. Among Markovian errors, there are three key types: incoherent (stochastic) errors, coherent errors, and other physical errors. Incoherent errors contain bit-flip error, phase-flip error and depolarizing error, *etc.* A bit flip changes a qubit's state from $|0\rangle$ to $|1\rangle$ or vice versa, while a phase flip alters the relative phase between the states without changing the probability amplitudes. when a depolarizing error occurs to a set of qubits, a random Pauli is applied to each qubit (Nielsen & Chuang, 2002). Coherent errors arise from unwanted or imperfect unitary rotations within quantum circuits (Huang et al., 2023). For slow noise processes, the error is often a coherent error(Beale et al., 2018). Other physical errors, such as amplitude damping, where energy dissipates into the environment, leading to state decay(Blume-Kohout et al., 2022).

*Non-Markovian* errors have memory effects, where the noise depends on the system's history. One signature of non-Markovian noise acting on a qubit can be obtained by analyzing the purity, $p$, of the qubit state, which is defined as $p = \text{Tr}[\rho^2]$. Oscillations of qubit coherence and purity over time indicate the presence of non-Markovian noise in idle qubits (Agarwal et al., 2024). The types of noise involved in this paper are detailed in Appendix A10.

## 2.3 GRAPH TRANSFORMER

Graph Transformers have emerged as a powerful tool for processing graph-structured data by combining the strengths of both Graph Neural Networks (GNNs) and Transformer architectures. To incorporate graph inductive biases into Transformers, many Graph Transformers integrate local message-passing mechanisms. For instance, some works incorporate sparse attention on local neighborhoods (Dwivedi & Bresson, 2020), or integrate various other types of Message-Passing Neural Networks (MPNNs) modules into their models (Chen et al., 2022). However, such graph Transformers may at least partially inherit some of the limitations of MPNNs, such as over-smoothing, over-squashing, and expressive power limitations (Xu et al., 2018). In contrast, Graph Transformers without (local) message-passing (Ma et al., 2023) allow each node to attend to other nodes, even if they are far apart in the graph structure. However, in this case, learning meaningful attention scores typically requires capturing important positional or structural relationships between nodes, and strong positional encodings are challenging to design for common graph data (Ma et al., 2023). Nevertheless, for the quantum error mitigation problem, our proposed method not only models quantum circuits as graphs to incorporate structural information but also generates effective positional encodings for each node based on the execution order of the quantum circuit to address this challenge.

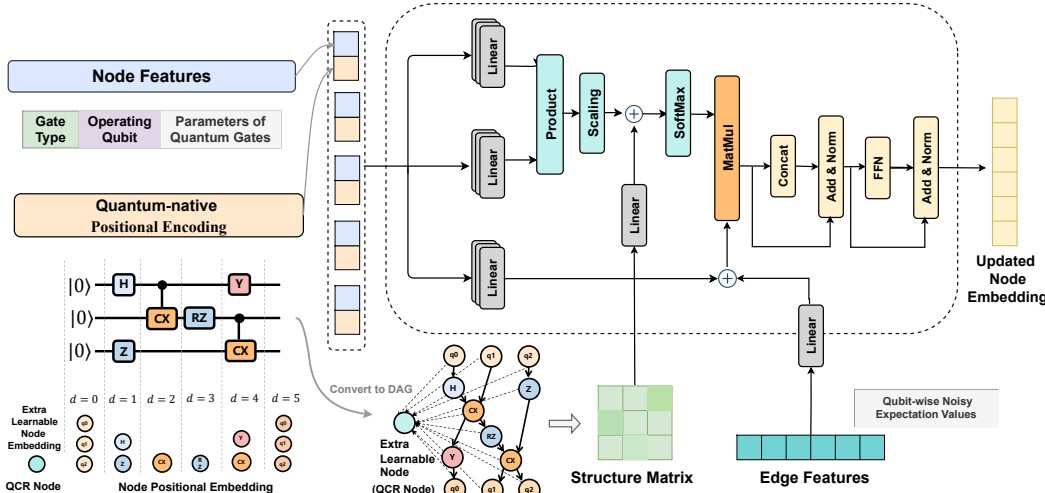

Figure 1: **Overview of the pipeline.** This figure depicts the architecture of the proposed quantum error mitigation method. **Left**: The circuit-to-graph encoding process transforms an input quantum circuit into a directed acyclic graph (DAG) to capture the circuit structure. Each operation is represented as a node with features including gate type, target qubit, and operation parameters, further enhanced by *Quantum-Native Positional Encoding* to capture the circuit's execution sequence. **Right**: The GTranQEM module processes the graph using multi-head attention mechanisms with a structure matrix and incorporates qubit-wise noisy measurement outcomes from edge features. This updates the node representations with global dependencies built in the quantum systems.

## 3 APPROACH

**Overview.** In this section, we present our method for quantum error mitigation in detail. The pipeline is displayed in Fig. 1. First, we introduce the problem formulation of the QEM adopted in this manuscript. Second, we propose a circuit-to-graph scheme tailored for the QEM task, which aggregates circuit information and dimensional noisy executions, with an additional quantum-native positional encoding incorporated to capture circuits' structural and temporal aspects. Finally, we present a non-message passing graph transformer GTranQEM to predict $\langle O \rangle^{\text{ideal}}$.

### 3.1 PROBLEM FORMULATION

We aim to predict the ideal expectation value $\langle O \rangle^{\text{ideal}}$ of a quantum circuit, formulated as a graph-level regression problem. Let $\mathcal{D} = \{(\mathcal{G}_k, \mathbf{X}_k, \mathbf{E}_k, y_k^{\text{noisy}}, y_k)\}_{k=1}^N$ be a dataset of $N$ samples, where each sample consists of a quantum circuit $C_k$ represented as a directed acyclic graph (DAG) $\mathcal{G}_k = (\mathcal{V}_k, \mathcal{E}_k)$, with node features $\mathbf{X}_k \in \mathbb{R}^{|\mathcal{V}_k| \times d}$ and edge features $\mathbf{E}_k \in \mathbb{R}^{|\mathcal{E}_k| \times d_e}$. The corresponding noisy and ideal expectation values are denoted by $y_k^{\text{noisy}} = \langle O \rangle_k^{\text{noisy}} \in \mathbb{R}$ and $y_k = \langle O \rangle_k^{\text{ideal}} \in \mathbb{R}$, respectively.

Our objective is to learn a function $f$, parameterized by a graph transformer model, that maps each sample to a prediction of its ideal expectation value which minimizes the mean squared error between the predicted and true ideal expectation values over the dataset:

$$\min_f \frac{1}{N} \sum_{k=1}^N (y_k - \hat{y}_k)^2, \quad \hat{y}_k = f\left(\mathcal{G}_k, \mathbf{X}_k, \mathbf{E}_k, y_k^{\text{noisy}}\right)$$

For simplicity, we omit the subscript $k$ in the following sections and refer to a single sample with graph $\mathcal{G} = (\mathcal{V}, \mathcal{E})$, node features $\mathbf{X}$, edge features $\mathbf{E}$, noisy expectation value $y^{\text{noisy}}$, and ideal expectation value $y$.

### 3.2 ENCODING SCHEME

We construct a directed acyclic graph $\mathcal{G} = (\mathcal{V}, \mathcal{E})$, where the node set $\mathcal{V}$ represents the quantum operations (gates), including start and end operations for each qubit. The edge set $\mathcal{E}$ encodes the directed qubit flow between operations.

**Node Features.** We denote $N = |\mathcal{V}|$ as the number of nodes. Each node $v_i \in \mathcal{V}$ is associated with a feature vector $\mathbf{x}'_i \in \mathbb{R}^{d'}$, $d'$ as the dimension of the node features. Specifically, the node feature vector $\mathbf{x}'_i$ is defined as:

$$\mathbf{x}'_i = [\mathbf{g}_i, \mathbf{q}_i, \theta_i],$$

where $\mathbf{g}_i \in \{0, 1\}^{N_{\text{gtype}}}$ is a one-hot encoding of the gate type, with $N_{\text{gtype}}$ being the total number of unique gate types in the dataset. $\mathbf{q}_i \in \{0, 1\}^{N_q}$ is a multi-hot encoding of the qubits that the gate operates on, with $N_q$ being the total number of qubits. $\theta_i \in \mathbb{R}$ is the normalized rotation angle for rotational gates (set to zero for non-rotational gates).

**Edge Features.** Each edge $(v_i, v_j) \in \mathcal{E}$ represents the flow of qubits from operation $v_i$ to operation $v_j$. The edge features $\mathbf{e}_{ij} \in \mathbb{R}^{d_e}$ capture the *qubit-wise noisy expectation values* of Pauli operators acting on the qubits carried by the edge. Formally, the edge connected to a two-qubit operation node acting on qubits $q_1$ and $q_2$ is defined as

$$\mathbf{e}_{ij} = \left[ \langle X_{q_1} \rangle^{\text{noisy}}, \langle Y_{q_1} \rangle^{\text{noisy}}, \langle Z_{q_1} \rangle^{\text{noisy}}, \langle X_{q_2} \rangle^{\text{noisy}}, \langle Y_{q_2} \rangle^{\text{noisy}}, \langle Z_{q_2} \rangle^{\text{noisy}} \right],$$

where $\langle \Omega_q \rangle^{\text{noisy}} = \langle \psi | I^{\otimes(q-1)} \otimes \Omega \otimes I^{\otimes(N_q-q)} | \psi \rangle^{\text{noisy}}, \Omega \in X, Y, Z$ denoting a Pauli operator on qubit $q$. If the edge carries only one qubit, the last three components are set to zero. Further details on the computation of qubit-wise expectation values are provided in Appendix A9.

This approach offers two main advantages. First, the method guarantees agnosticism to the noise types as it encodes noisy expectation values with no explicit knowledge of the noise type or error rates required. Second, compared to the previous method (Liao et al., 2024), which aggregates only the circuit-level noisy expectation value $\langle Z^{\otimes N_q} \rangle^{\text{noisy}}$ as a scalar global feature, our approach includes multi-dimensional qubit-level noisy information, potentially enhancing GTranQEM robustness.

### 3.3 QUANTUM-NATIVE GRAPH TRANSFORMER

#### 3.3.1 QUANTUM-NATIVE POSITIONAL ENCODING

Graphs, though mapping quantum circuits structure naturally, inherently lack sequential order. On the other hand, in quantum circuits, gates are applied sequentially along each qubit and may act concurrently across different qubits. To capture this temporality in addition to the structure, we introduce a quantum-native positional encoding scheme.

Specifically, for each node $v_i \in \mathcal{V}$, we assign a positional encoding $\mathbf{p}_i \in \mathbb{R}^{d_p}$ that reflects both the layer at which the gate operates and the qubits it acts upon. Specifically, let $L$ denote the total number of quantum layers in circuit $C$. We assign a layer index $l_i \in \{0, 1, \ldots, L+1\}$ of each node $v_i$, where $l_i = 0$ corresponds to the start operation, $l_i = L+1$ corresponds to the end operation. Let $Q_i \subseteq \{1, 2, \ldots, N_q\}$ denote the set of qubits that gate $v_i$ acts upon. For single-qubit gates, $Q_i = \{q\}$; for two-qubit gates, $Q_i = \{q_1, q_2\}$.

We define the positional encoding $\mathbf{p}_i$ by first flattening the pair $(l_i, Q_i)$ to obtain a unique 1D temporal position index of each $v_i$, and then applying the positional encoding scheme from Vaswani et al. (2017) to obtain a vector representation. As $\mathbf{p}_i$ captures the execution order (via $l_i$) and the qubit interactions (via $Q_i$) of each node, it allows the model to leverage both the intrinsic properties of the quantum operations, the structural context within the circuit, and the related temporal information, enabling more accurate modeling of the entire quantum systems.

#### 3.3.2 GTRANQEM ARCHITECTURE

With the quantum-native positional encoding, the final node features are then given by concatenation:

$$\mathbf{x}_i = [\mathbf{x}'_i \,||\, \mathbf{p}_i] \in \mathbb{R}^d, \quad d = d' + d_p.$$

This enriched node feature matrix $\mathbf{X} \in \mathbb{R}^{N \times d}$ and edge feature matrix $\mathbf{E}$ together are utilized as input for the GTranQEM layer, as displayed in Fig. 1. Extending the standard Transformer encoder (Vaswani et al., 2017), each layer consists of *multi-head self-attention mechanisms* with *global attention bias*, followed by feed-forward networks (FFNs) for nodes, with layer normalization and residual connections (Xiong et al., 2020).

**Multi-Head Attention.** Specifically, given node features $\mathbf{x}_i^\ell \in \mathbb{R}^d$ and edge features $\mathbf{e}_{ij}^\ell \in \mathbb{R}^{d_e}$ at layer $\ell$, the attention scores are computed as:

$$a_{ij}^{k,\ell} = \left( \frac{(\mathbf{Q}^{k,\ell}\mathbf{x}_i^\ell)^\top (\mathbf{K}^{k,\ell}\mathbf{x}_j^\ell)}{\sqrt{d_k}} + \beta M_{ij} \right), i, j \in \mathcal{V}, \tag{1}$$

for attention head $k \in \{1, \dots, H\}$, where $\mathbf{Q}^{k,\ell} \in \mathbb{R}^{d_k \times d}$ represents the Query vector from the central node $i$, and $\mathbf{K}^{k,\ell} \in \mathbb{R}^{d_k \times d}$ represents the Key vector from **all** nodes in the graph instead of neighbor nodes. $d_k = d/H$ as in (Vaswani et al., 2017), and $\beta$ is a learnable scalar parameter for the global attention bias $M_{ij}$ (we shall introduce with details in the following section). The attention weights are accordingly obtained via the softmax function:

$$\alpha_{ij}^{k,\ell} = \mathrm{softmax}_{j \in \mathcal{V}} \left( a_{ij}^{k,\ell} \right). \tag{2}$$

For edge features, we first update $\mathbf{E}$ via $\mathbf{e}_{ij}^{\ell+1} = \sigma(\mathbf{W}_e \mathbf{e}_{ij}^\ell + \mathbf{b}_e), (v_i, v_j) \in \mathcal{E}$, where $\mathbf{W}_e \in \mathbb{R}^{d \times d_e}$ and $\mathbf{b}_e$ denotes a projection weight matrix and bias. $\sigma(\cdot)$ denotes an activation function. To employ qubit-wise noisy measurement outcomes built-in $\mathbf{E}$ to assist the ideal expectation value prediction, we browse $\mathcal{G}$ to incorporate egde features towards its source nodes. Namely, the graph-wise edge attributes for node $j$ is:

$$\overline{\mathbf{e}}_j^{\ell+1} = \frac{1}{D_j} \sum_{s|(v_j,v_s)\in\mathcal{E}} \mathbf{e}_{js}^{\ell+1}, \tag{3}$$

where $D_j$ denotes the degree of node $j$. The averaged edge feature $\overline{\mathbf{e}}_j^{\ell+1}$ represents the mean of all edge features connected to node $j$, which incorporates global edge information into each node's updated feature. Accordingly, node features are then updated as follows:

$$\begin{aligned}
\hat{\mathbf{x}}_i^{\ell+1} &= \mathbf{O}_h^\ell \left[ \tilde{\mathbf{x}}_i^{1,\ell+1} \| \dots \| \tilde{\mathbf{x}}_i^{H,\ell+1} \right], \\
\tilde{\mathbf{x}}_i^{k,\ell+1} &= \sum_{j \in \mathcal{V}} \alpha_{ij}^{k,\ell} \left( \mathbf{V}^{k,\ell}\mathbf{x}_j^\ell + \overline{\mathbf{e}}_j^{\ell+1} \right),
\end{aligned} \tag{4}$$

with $\mathbf{V}^{k,\ell} \in \mathbb{R}^{d_k \times d}$, denotes the value vectors from all nodes, $\mathbf{O}_h^\ell \in \mathbb{R}^{d \times (Hd_k)}$ represents a projection matrix, and $\|$ denoting the concatenation operation.

Overall, this approach enables the model to integrate both node and edge information without relying on explicit message passing along the adjacency structure.

**Global Attention Bias.** Instead of relying on message passing, it is important to find a way to include the graph topology for learning better representations. Previous studies demonstrate that structural information can be incorporated into attention score as a *attention bias* to guide the graph transformers without traditional message passing (Rampášek et al., 2023; Xia et al., 2023). Therefore, we compute a *structure matrix* $M \in \mathbb{R}^{N \times N}$ for the graph $\mathcal{G}$, where $M_{ij}$ represents the shortest distance between nodes $v_i$ and $v_j$ derived from the Floyd-Warshall algorithm (Floyd, 1962). Namely,

$$M_{ij} = \begin{cases} \mathrm{distance}(v_i, v_j) & \text{if } \mathrm{distance}(v_i, v_j) \leq M_{\max} \\ M_{\max} & \text{otherwise,} \end{cases}$$

where $M_{\max}$ denotes a maximum allowed step count.

The structure matrix $M$ is accumulated to the attention score as a bias with a learnable scalar weight in Eq. 1. Given the prevalence of quantum entanglement in most circuits, $M$ enables GTranQEM to grasp the global dependencies rather than focusing solely on local neighborhoods. Overall, this mechanism is adaptive to QEM settings and can enhance the global awareness and the expressive power of graph transformers for the QEM problem.

The node representations will eventually pass through separate **Feed-Forward Networks** with residual connections and layer normalization, followed Vaswani et al. (2017).

**Quantum Circuit-Representative Node.** To enhance the model's capacity to capture global patterns and dependencies within quantum circuits, we introduce a virtual *Quantum Circuit-Representative Node* (QCR node) into the graph-level regression model, which is a learnable node to capture the whole circuit representation, as shown in Fig. 1. The QCR node is designed to aggregate information from all other nodes via the attention mechanism, thereby preserving crucial information that might be lost through mean or max pooling operations.

Formally, we augment the node set $\mathcal{V}$ by adding the QCR node $v_{\text{QCR}}$, resulting in an extended node set $\mathcal{V}' = \mathcal{V} \cup \{v_{\text{QCR}}\}$. The QCR node is connected to all other nodes by special directed edges:

$$\mathcal{E}' = \mathcal{E} \cup \{(v_i, v_{\text{QCR}}) \mid v_i \in \mathcal{V}\}.$$

There are no outgoing edges from the QCR node, and the feature vector of the QCR node, $\mathbf{x}_{\text{QCR}} \in \mathbb{R}^d$, is initialized as a learnable parameter.

During the transformer layers, the QCR node participates in the self-attention mechanism, attending to all other nodes and aggregating information from the entire graph with the qubit-wise noisy information from edges. Eventually, we utilize the $\mathbf{x}_{\text{QCR}}^L$ of the QCR node as the input to the subsequent regression module (error mitigator) rather than applying a pooling operation overall nodes.

### 3.4 ERROR MITIGATOR

After passing through $L$ GTranQEM layers, we concatenate the representation of QCR node, $\mathbf{x}_{\text{QCR}}^L \in \mathbb{R}^d$ with the noisy expectation value ($y_{\text{noisy}}$) obtained from the quantum circuit execution to form the input vector:

$$\mathbf{z} = [\,\mathbf{x}_{\text{QCR}}^L \,\|\, y_{\text{noisy}}\,].$$

This vector $\mathbf{z}$ is then fed into a regression module consisting of two fully connected layers with a non-linear activation function in between. The regression module outputs the predicted noise-mitigated expectation value $\hat{y} \in \mathbb{R}$.

## 4 EXPERIMENTS

In this section, we first propose the data augmentation schemes that refine the current framework of machine learning-based QEM. We then elaborate on the experiment settings we utilized and display the performance of GTranQEM.

### 4.1 DATA AUGMENTATION STRATEGIES

#### 4.1.1 REDUCING DEPENDENCE ON IDEAL EXPECTATION VALUES FOR TRAINING

As discussed in the introduction, a primary challenge in training learning-based QEM schemes is reliance on ideal expectation values (EVs) as labels for training data, potentially limiting ML-based models' capability. To address this issue, we reduce the need for direct reliance on ideal EVs by constructing training circuits with known ideal outputs through **inversed circuit composition**. Specifically, for each circuit $C$ in the training set, we generate an augmented circuit $\bar{C}$ by appending its inverse $C^{-1}$ to itself, resulting in the composite circuit: $\bar{C}_{\text{train}} = C \cdot C^{-1}$.

We initialize the circuit with a random state $|\psi_{0,k}\rangle$. In the absence of noise, the composite circuit $\bar{C}$ satisfies $UU^{-1} = I$, leaving the state unchanged. Consequently, the ideal expectation value of the observable $Z^{\otimes n}$ (the Pauli-$Z$ operator applied to all $n$ qubits) serves as the training label is:

$$y_k^{\text{ideal}} = \langle\psi_{0,k}|Z^{\otimes n}|\psi_{0,k}\rangle. \tag{5}$$

It allows to obtain ideal expectation values without executing the original circuits. Moreover, the augmented circuits retain a structure similar to the original circuits, incorporating the same types of gates. Such similarity ensures that the training data encapsulates noise characteristics comparable to those of the original test circuits.

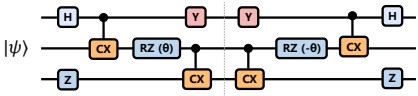

Figure 2: Visualization of the inversed circuit composition. The left half: original circuits.

Formally, the training dataset is constructed as: $\mathcal{D}^{\text{train}} = \left\{\left(\bar{\mathcal{G}}_k, \bar{y}_k^{\text{noisy}}, \bar{y}_k^{\text{ideal}}\right)\right\}_{k=1}^N$, where each augmented circuit

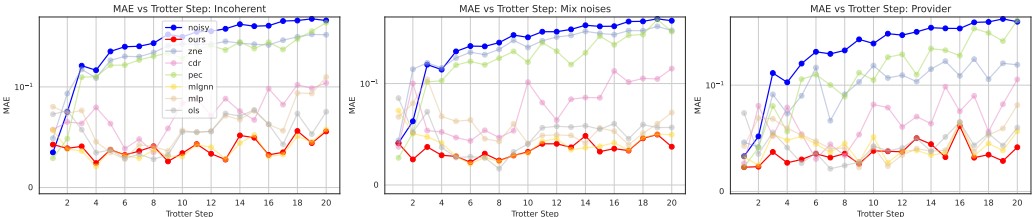

Figure 3: Visualization of Mitigation Performance via Trotter steps under three noise settings for 6-qubit Trotterized circuits. The symlog scale is used in the y-axis for clearer visualization.

| Method | All-6-Trotter | | | | | | Max-6-Random | | | | | |
| --- | --- | --- | --- | --- | --- | --- | --- | --- | --- | --- | --- | --- |
| | Incoherent | | Mixed Noise | | Fake Provider | | Incoherent | | Mixed Noise | | Fake Provider | |
| | MAE↓ | CIR↑ | MAE↓ | CIR↑ | MAE↓ | CIR↑ | MAE↓ | CIR↑ | MAE↓ | CIR↑ | MAE | CIR↑ |
| Raw Noisy | 0.3574±0.2026 | - | 0.3227±0.1811 | - | 0.2911±0.1697 | - | 0.0737±0.0750 | - | 0.0721±0.0733 | - | 0.1405±0.1398 | - |
| ZNE | 0.2558±0.1397 | 0.8067 | 0.2746±0.1544 | 0.7900 | 0.1181±0.1071 | 0.8767 | 0.0647±0.0826 | 0.6667 | 0.0684±0.0772 | 0.5800 | 0.1102±0.1190 | 0.5900 |
| CDR | 0.0740±0.0776 | 0.8700 | 0.0803±0.0774 | 0.8967 | 0.0663±0.0964 | 0.8733 | 0.0857±0.1641 | 0.5667 | 0.0743±0.1412 | 0.6000 | 0.0966±0.1764 | 0.6000 |
| PEC | 0.2376±0.2060 | 0.7900 | 0.2203±0.2096 | 0.7633 | 0.1801±0.2048 | 0.8033 | 0.0461±0.0423 | 0.6367 | 0.0526±0.0501 | 0.6200 | 0.1461±0.1482 | 0.4333 |
| OLS | 0.0548±0.0516 | 0.8667 | 0.0491±0.0430 | 0.8600 | 0.0435±0.0405 | 0.8667 | 0.0335±0.0339 | 0.6800 | 0.0344±0.0325 | 0.6833 | 0.0467±0.0328 | 0.6833 |
| MLP | 0.0639±0.0515 | 0.8600 | 0.0581±0.0518 | 0.8667 | 0.0479±0.0445 | 0.8500 | 0.0291±0.0258 | 0.7300 | 0.0289±0.0256 | 0.7300 | 0.0447±0.0306 | 0.6933 |
| GNN | 0.0388±0.0335 | 0.8900 | 0.0397±0.0318 | 0.8500 | 0.0415±0.0389 | 0.8767 | 0.0325±0.0323 | 0.6833 | 0.0351±0.0315 | 0.6933 | 0.0472±0.0328 | 0.6733 |
| GTranQEM | 0.03610±0.0310 | 0.9067 | 0.0315±0.0272 | 0.9267 | 0.0371±0.0292 | 0.9133 | 0.0267±0.0228 | 0.7700 | 0.0290±0.0235 | 0.7467 | 0.0440±0.0338 | 0.7433 |

Table 1: Mean Absolute Error (MAE) and Circuit Improvement Ratio (CIR) for different methods under various noise settings and circuit configurations. The best values in each column are highlighted in red, and the second-best values are highlighted in blue. Formulas for the metric calculation are included in Appendix A12.2.

$\bar{C}_k = C_k \cdot C_k^{-1}$ corresponds to a graph $\bar{\mathcal{G}}_k$, and $\bar{y}_k^{\text{ideal}} = \langle \psi_{0,k} | Z^{\otimes n} | \psi_{0,k} \rangle$. The test dataset consists of the original circuits, defined as: $\mathcal{D}^{\text{test}} = \{\mathcal{G}_k\}_{k=1}^{M}$. This strategy enables the generation of training data for large-qubit circuits without the computational burden of simulating their ideal EVs, thereby effectively allowing the model to learn the underlying noise characteristics.

### 4.1.2 Noise Scaling Amplification

To enhance the model's robustness to varying noise levels, we adopt a noise amplification technique inspired by ZNE (Temme et al., 2017; Li & Benjamin, 2017). Specifically, we scale the noise levels by factors $\lambda$ sampled from the interval $[1, \lambda_{\max}]$. For each scaling factor $\lambda$, the circuit is executed under the scaled noise model to obtain the corresponding noisy expectation value $y_k^{\lambda,\text{noisy}}$ and qubit-wise expectation values used as edge features. This augmentation introduces diverse noise profiles into the dataset, enabling the model to generalize across different noise levels and enhancing its robustness to noise variability. Formally, for each circuit $C_k$ in the training set, we generate augmented data points:

$$\left\{ \left( \mathcal{G}_k^\lambda, y_k^{\lambda,\text{noisy}}, y_k^{\text{ideal}} \right) \,\middle|\, \lambda \in \Lambda \right\}, \tag{6}$$

where $\Lambda = \{\lambda_1, \lambda_2, \ldots, \lambda_L\}$ represents the set of noise scaling factors.

By integrating these data augmentation strategies, we reduce the dependence on ideal expectation values for training and mitigate the model's sensitivity to noise with high variance, thereby better reflecting the conditions of real quantum devices.

### 4.2 Setups and Competitors

Experiment setups, computation resources, and model configurations are displayed in the Appendix A12.1. Plenty of quantum error mitigation techniques have been developed for various circuit types and noise models (Cai et al., 2023). In this manuscript, we select baselines that are not only proven to be highly effective but also capable of handling different circuit types and noise models, similar to GTranQEM. Specifically, for classical-quantum methods, we use **Zero-Noise Extrapolation (ZNE)** (Temme et al., 2017; Li & Benjamin, 2017) and **Probabilistic Error Cancellation** (Van den Berg et al., 2023). Additionally, we include learning-based methods such as **ML-GNN** (Liao et al., 2024), **Clifford Data Regression (CDR)** (Czarnik et al., 2021), **OLS** (Liao et al., 2024), and **MLP** (Kim et al., 2020). Further details can be found in Appendix A11.

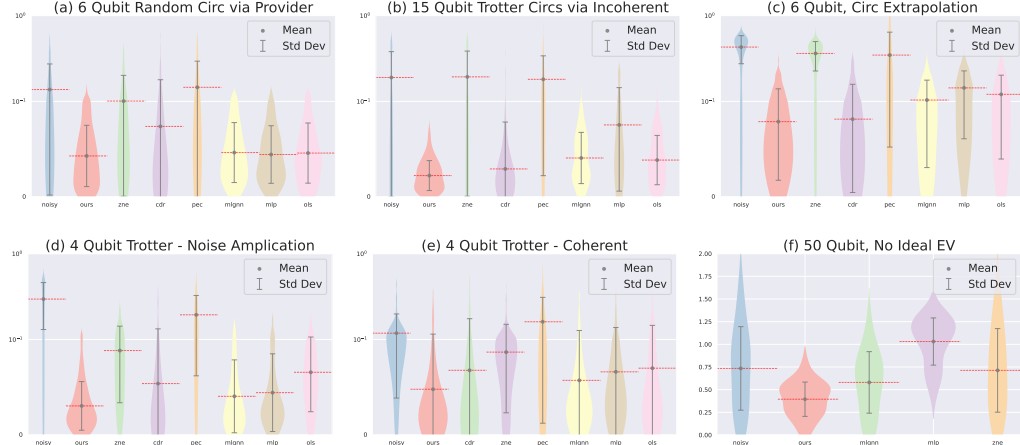

Figure 4: Performance of mitigation methods across various circuit configurations and noise settings. The x-axis displays different QEM methods, while the y-axis represents the absolute error distribution. Panels (a)–(e) use a symlog scale for clearer visualization, as error rates are low after mitigation. Each panel title indicates the specific setting. The width of each violin plot reflects data density at each value, with a broader shape near zero indicating better performance. Our method is shown in red, and raw noisy data is in blue.

### 4.3 QUANTUM CIRCUITS

The experiments mainly focus on two types of circuits: the Max-N-Random circuits and the All-N-Trotter circuits. Details for circuit and dataset configurations are provided in Appendix A12.4.

**Max-Qubit random circuits:** We utilize the random-structure circuits to demonstrate the generality of GTranQEM. In this setting, circuits in a dataset consist of at most $N_q$ qubits and most $N_{gate}$. The allowed gates are chosen from {Pauli-X, Pauli-Y, Pauli-Z, Hadamard, RX, RY, RZ, Controlled-NOT, Controlled-Z, SWAP, RZZ}. The types of, the orders of, and the qubits being operated by quantum gates are randomized within each circuit.

**Trotterized circuits:** To benchmark our protocol's performance on structured circuits, we employ Trotterized brickwork circuits following the approach of Liao et al. (2024). We focus on the Trotterized dynamics of the 1-D transverse-field Ising model (TFIM) as described in Chakrabarti et al. (1996).

### 4.4 QUANTUM ERROR SETTINGS

We evaluate the effectiveness of GTranQEM using four types of noise models. (1) **IBM's fake providers**: We simulate noisy quantum circuit executions using several IBM fake providers, such as FakeHanoiV2, which mimic the complex noise profiles of real quantum devices. This demonstrates the capability of GTranQEM in handling realistic quantum noise. (2) **Incoherent noise setting**: We construct a noise model with depolarization errors, which degrade the quantum unitary state into a mixed state, to showcase the efficacy of GTranQEM in addressing a specific noise type. (3) **Mixed noise setting**: This model combines Pauli X errors, depolarization errors, and readout errors, aiming to evaluate GTranQEM under a customized, complex mixed-error scenario. (4) **Coherent noise setting:** This model combines a coherent over-rotation with the Fake Lima Provider. (5) **IBM's Brisbane** We execute the 50-qubit circuits on IBM's Brisbane quantum device and obtain noisy expectation values. Configurations of each error setting are presented in Appendix A12.3.3.

### 4.5 EXPERIMENTAL PERFORMANCE

**Performance over Standard Settings.** *Trotterized Circuits.* We first evaluated GTranQEM on 6-qubit Trotterized circuits with Trotter steps ranging from 1 to 20 with three different noises. As depicted in Fig. 3, GTranQEM consistently outperforms existing QEM techniques, maintaining lower MAE values across all Trotter steps, which highlights the robustness of our approach in mitigating errors as circuit depth increases. Tab. 1 demonstrates that GTranQEM is the only method that consistently achieves a circuit improvement ratio (CIR) over $90\%$. For 15-qubit circuits with incoherent noise,

| Method | Circuit-Scale Extrapolation | | Noise-Scale Amplification | | 15-Qubit-Circuit | | Coherent Noise | |
|---|---|---|---|---|---|---|---|---|
| | MAE ↓ | CIR ↑ | MAE ↓ | CIR ↑ | MAE ↓ | CIR ↑ | MAE ↓ | CIR ↑ |
| Raw Noisy | 0.4294±0.1538 | - | 0.2958±0.1646 | - | 0.1900±0.1891 | - | 0.1324±0.1238 | - |
| ZNE (Li & Benjamin, 2017) | 0.3621±0.1352 | 0.9300 | 0.0906±0.0620 | 0.9050 | 0.1933±0.1917 | 0.4700 | 0.0974±0.1088 | 0.8667 |
| CDR (Czarnik et al., 2021) | 0.0814±0.0765 | 0.9500 | 0.0669±0.1036 | 0.8900 | 0.0488±0.1263 | 0.7000 | 0.0826±0.1564 | 0.7933 |
| PEC (Van den Berg et al., 2023) | 0.3464±0.2928 | 0.6000 | 0.1986±0.1388 | 0.7600 | 0.2076±0.2535 | 0.4100 | 0.1756±0.1827 | 0.4467 |
| MLGNN (Liao et al., 2024) | 0.1038±0.0727 | 0.9300 | 0.0425±0.0431 | 0.8700 | 0.0421±0.0299 | 0.6300 | 0.0699±0.1019 | 0.8167 |
| OLS (Liao et al., 2024) | 0.1209±0.0809 | 0.9500 | 0.0696±0.0493 | 0.8550 | 0.0411±0.0312 | 0.6400 | 0.0813±0.1082 | 0.7833 |
| MLP (Kim et al., 2020) | 0.1438±0.0825 | 0.9600 | 0.0466±0.0462 | 0.8450 | 0.0754±0.0691 | 0.6700 | 0.0779±0.1012 | 0.8467 |
| GTranQEM | 0.0788±0.0611 | 0.9700 | 0.0325±0.0304 | 0.8950 | 0.0246±0.0230 | 0.6800 | 0.0619±0.1017 | 0.8533 |

Table 2: Mean Absolute Error (MAE) and Circuit Improvement Ratio (CIR) for different methods under four advanced settings. Formulas for the metric calculation are included in Appendix A12.2.

our method achieves an MAE of $0.0245$, significantly smaller than other baselines, with the smallest standard deviation (std) of $0.0245$, as shown in Tab. 2. The error distribution of the 15-qubit circuits is displayed in Fig. 4(b), where our method has the most significant concentration towards zero, showcasing its effectiveness in handling larger circuit configurations.

*Random Circuits.* Superior was also observed for random circuits. As presented in Tab. 1, the MAE and CIR indicate that GTranQEM achieves a noticeable error drop compared to baselines. For example, under incoherent noise, our approach yields an MAE of $0.0267$, whereas the second-best GNN method achieves $0.0291$. Fig. 4(a) visualizes the error distribution under provider noise, illustrating that our method greatly outperforms traditional QEM methods and also has notable improvement among other ML-based methods.

**Performance in Alternative Task Settings.** Additionally, we evaluated our method under *noise amplification*, *circuit extrapolation*, and *coherent noise* conditions. As shown in Tab. 2 and Figs. 4 (c)-(e), GTranQEM consistently outperforms other mitigation strategies across these scenarios. For example, in the circuit-scale extrapolation task—where QEM methods are trained on shallow circuits with fewer than 15 Trotter steps and tested on deeper circuits with 16 to 20 Trotter steps—GTranQEM achieves an MAE of $0.0325$ and a CIR of $0.895$. These results demonstrate the robustness and generality of GTranQEM across various tasks and noise settings.

**Performance with Refined Learning Paradigm.** We further evaluated GTranQEM in scenarios where ideal expectation values of original training circuits are excluded through inverse circuit composition. Two experiments were conducted: one using 6-qubit Trotter circuits subjected to mixed noise and another involving large-scale 50-qubit circuits executed on a real quantum device (IBM Brisbane). As shown in Figs. 4 (f) and 7, our method significantly outperforms baseline mitigation strategies and achieves a substantial reduction in error compared to raw noisy data. The error rates are notably reduced towards zero, demonstrating the effectiveness of GTranQEM and our refined learning paradigm (see Appendix A12.4.2 for further analysis).

## 5 CONCLUSION

We have devised a non-message-passing graph transformer, GTranQEM, for quantum error mitigation that effectively captures both the graph structure of quantum circuits and the global dependencies inherent in quantum systems. It takes quantum circuit graphs with built-in qubit-wise noisy information as input, generated through our devised encoding scheme. Unlike current machine learning-based methods, our graph transformer aggregates information across the entire graph using a structure matrix to guide the graph topology, incorporates a quantum-native positional encoding, and aggregates graph representations within a global context into a QCR node. The representation of the QCR node, which encapsulates both the circuit's structure and the global dependencies, is then employed as input for the error mitigator, thereby reducing the impact of errors in circuit executions. The source code will be available at: https://github.com/btyll/GTranQEM.

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

## A6 Broader Impact

Our proposed method achieves nonlocal quantum error mitigation while preserving circuit topology through a non-message-passing graph transformer. By modeling quantum circuits as graphs and employing a transformer architecture, we capture both their structural topology and global dependencies, such as Bell nonlocality (Bell, 1964; Brunner et al., 2014). This approach addresses limitations in existing machine learning-based error mitigation methods (Kim et al., 2020; Liao et al., 2024) that often overlook circuit structure or nonlocal characteristics. Enhancing the reliability of quantum computations in the NISQ era (Brooks, 2019), our work contributes to advanced quantum algorithms and error mitigation strategies that fully exploit circuit topology and global entanglement. Ultimately, this advancement promotes practical quantum advantage and accelerates the integration of quantum computing into diverse scientific and industrial applications.

## A7 Limitation

Compared to the traditional QEM methods, the ML-based QEM methods usually require the acquisition of the training set and training process. Offline training introduces additional time costs, but the advantage is the ability to quickly infer unseen samples. On the other hand, our method uses measurement information from the circuit to construct the edge features, which may increase the measurement cost during circuit execution. However, fortunately, in our circuit, we only use the results of Pauli measurements on each qubit, which are considered local observables. By employing the classical shadow algorithm (Huang et al., 2020), we can reduce the original $M$ measurements to only $log(M)$ measurements, and this reduction is independent of the number of qubits in the quantum circuit. This approach allows us to address potential scalability issues as circuit sizes grow.

## A8 Discussion

We want to emphasize the key innovations of our paper lie in the targeted, quantum-representative design of GTranQEM, including the following aspects:

1) **Novel Encoding Scheme:** We introduce a unique encoding approach for QEM. Following the standard approaches in graph learning for quantum computing (Wang et al., 2022b; Moflic et al., 2023; He et al., 2023), quantum circuits are represented as DAGs, mapping each gate in the circuit to a node in the graph. Meanwhile, several works are designed to utilize quantum circuits to learn graph data (Yan et al., 2023; Wu et al., 2024b), and these fall within the realm of quantum machine learning. However, our encoding scheme differs from previous work in several key aspects including node features with circuit attributes as well as quantum topological positional encoding, and edge features with noisy qubit-wise measurement outcomes 3.2. Overall, our encoding captures both the circuit topology and the measurement information in a unified framework, strengthening the model's ability to perform mitigation tasks in complex quantum systems.

3) **Customized Quantum-Native Non-Message Passing Graph Transformers**: We are among the first to apply a non-message passing graph transformer architecture to the QEM problem. Traditional message-passing mechanisms focus on neighborhood aggregation, failing to capture the long-range dependencies within the quantum system. In contrast, GTraQEM leverages the circuit's topology incorporated in the graph encoding, the non-message passing aggregation scheme, and a learnable QCR node to further facilitate graph-level feature aggregation, making GTraQEM particularly quantum-native and customized for QEM tasks.

4) **Refinement of the Learning-Based QEM Paradigm**: Previous learning-based QEM methods (Kim et al., 2020; Liao et al., 2024) often rely on obtaining ideal expectation values of training circuits through classical simulation for mitigator tuning, which becomes impractical for larger circuits due to the exponential scaling of computational resources. We propose to refine the paradigm by constructing the training set with standard circuits composed with their inverse with random initial state assigned. This approach allows us to obtain ideal EVs as labels of the training circuits for supervision without circuit executions, as the initial state becomes the final quantum state. This significantly reduces ML-based methods' reliance on the simulation of ideal EVs, achieving label-efficient training.

5) **Comprehensive ML Perspective and Effective Pipeline**: Given the limited number of existing ML-based QEM methods, our work provides a fresh perspective by thoroughly exploring the intersection of Graph learning and QEM. We propose an effective and thorough pipeline for error mitigation across various circuit types and scales, highlighting its potential for wide applicability in the quantum computing community.

## A9   EDGE FEATURE CONSTRUCTION

**1. Construction of Pauli-Identity Operators for the Qubit.**   The Pauli-Identity operator $\Omega_q$ acting on the $q$-th qubit in an $N_q$-qubit system is constructed as:

$$\Omega_q = I^{\otimes(q-1)} \otimes P_q \otimes I^{\otimes(N_q-q)},$$

where:

- $I$ is the $2 \times 2$ identity matrix.
- $P_q \in \{X_q, Y_q, Z_q\}$ is one of the Pauli matrices acting on qubit $q$.
- $I^{\otimes n}$ denotes the $n$-fold tensor product of the identity operator.

**2. Qubit-wise Noisy Expectation Values**

- For each qubit $q$ carried by an edge $(v_i, v_j)$, we compute the noisy expectation values of the set of the Pauli-Identity operators mentioned previously.
- The qubit-wise noisy expectation value for a Pauli operator $\Omega_q$ acting on qubit $q$ is defined as:

$$\langle \Omega_q \rangle^{\text{noisy}} = \text{Tr}\left[\rho^{\text{noisy}} P_q\right],$$

where $\rho^{\text{noisy}}$ is the noisy state vector of the system.

**3. Computation of Noisy Expectation Values**

- For each qubit $q$ on the edge $(v_i, v_j)$, compute:

$$\langle X_q \rangle^{\text{noisy}} = \text{Tr}\left[\rho^{\text{noisy}} \left(I^{\otimes(q-1)} \otimes X \otimes I^{\otimes(N_q-q)}\right)\right],$$

$$\langle Y_q \rangle^{\text{noisy}} = \text{Tr}\left[\rho^{\text{noisy}} \left(I^{\otimes(q-1)} \otimes Y \otimes I^{\otimes(N_q-q)}\right)\right],$$

$$\langle Z_q \rangle^{\text{noisy}} = \text{Tr}\left[\rho^{\text{noisy}} \left(I^{\otimes(q-1)} \otimes Z \otimes I^{\otimes(N_q-q)}\right)\right].$$

**4. Edge Feature Vector Assembly**

- **Single-Qubit Edge**:
  - If the edge carries only one qubit $q$, the edge feature vector $\mathbf{e}_{ij} \in \mathbb{R}^6$ is constructed as:

  $$\mathbf{e}_{ij} = \left[\langle X_q \rangle^{\text{noisy}}, \langle Y_q \rangle^{\text{noisy}}, \langle Z_q \rangle^{\text{noisy}}, 0, 0, 0\right].$$

  - The last three components are set to zero to maintain a consistent feature dimension.
- **Two-Qubit Edge**:
  - If the edge carries two qubits $q_1$ and $q_2$, the edge feature vector $\mathbf{e}_{ij} \in \mathbb{R}^6$ is constructed as:

  $$\mathbf{e}_{ij} = \left[\langle X_{q_1} \rangle^{\text{noisy}}, \langle Y_{q_1} \rangle^{\text{noisy}}, \langle Z_{q_1} \rangle^{\text{noisy}}, \langle X_{q_2} \rangle^{\text{noisy}}, \langle Y_{q_2} \rangle^{\text{noisy}}, \langle Z_{q_2} \rangle^{\text{noisy}}\right].$$

By following these steps, we construct edge features that capture essential quantum state information encoded in the noisy expectation values of Pauli operators for the qubits flowing between quantum operations. This representation is crucial for analyzing quantum circuits using graph-based machine learning techniques.

## A10 QUANTUM ERRORS

Quantum error sources are a critical challenge in quantum computing. These errors, ranging from coherent to incoherent and systematic to random, arise from imperfect qubit control, unwanted interactions, and measurement inaccuracies. In this paper, we conduct QEM experiments on incoherent errors, coherent errors, and readout errors.

### A10.1 INCOHERENT ERRORS

Incoherent errors are caused by random interactions with the environment, leading to non-unitary dynamics. These errors cause decoherence and loss of quantum information. Incoherent errors (from entanglement of the physical qubits with a memoryless environment) can be modeled as random Pauli operators on the physical qubits. In our paper, we involve two types of incoherent errors: bit-flip error and depolarizing error.

A bit-flip channel transforms the quantum state from $|0\rangle$ to $|1\rangle$ (or vice versa) with a probability of $1 - p$, and it has the following operators:

$$E_0 = \sqrt{p}I = \sqrt{p} \begin{bmatrix} 1 & 0 \\ 0 & 1 \end{bmatrix}, \tag{7}$$

$$E_1 = \sqrt{1-p}X = \sqrt{1-p} \begin{bmatrix} 0 & 1 \\ 1 & 0 \end{bmatrix}. \tag{8}$$

Its effect on a state $\rho$ is given by

$$\Phi_{BF}[\rho] = p\rho + (1-p)X\rho X. \tag{9}$$

The depolarizing channel is an important type of quantum noise. Imagine that we take a single qubit and depolarize it with probability $p$, meaning the qubit is replaced by a completely mixed state $I/2$. The single qubit remains unchanged with probability $1 - p$. The state of the quantum system $\rho$ after this noise is given by:

$$\Phi_{DE}[\rho] = \frac{pI}{2} + (1-p)\rho. \tag{10}$$

If we observe that for any $\rho$, the following holds:

$$\frac{I}{2} = \frac{\rho + X\rho X + Y\rho Y + Z\rho Z}{4},$$

then we can obtain:

$$\Phi_{DE}[\rho] = \left(1 - \frac{3p}{4}\right)\rho + \frac{p}{4}(X\rho X + Y\rho Y + Z\rho Z). \tag{11}$$

This shows that the depolarizing channel has the set of operators $\{\sqrt{1 - 3p/4}I, \sqrt{p}X/2, \sqrt{p}Y/2, \sqrt{p}Z/2\}$.

### A10.2 COHERENT ERRORS

Coherent errors arise from unwanted or imperfect unitary rotations within quantum circuits, which can be modeled as:

$$U(\theta) = e^{-\frac{i}{2}\theta\sigma}, \tag{12}$$

where $\theta = (\theta_1, ..., \theta_{4^{N_q}})$ represents the magnitude of the coherent error on the $4^{N_q}$ Pauli basis. These errors transform noiseless pure states into other pure states while preserving quantum coherence due to the unitary nature of the operations. Although they are purity-preserving, coherent errors can significantly undermine the reliability of multi-qubit quantum computations.

### A10.3 READOUT ERRORS

Readout errors occur during the measurement process of quantum states. These errors manifest when the outcome of a quantum measurement does not accurately reflect the actual state of the qubit. For instance, a qubit in the $|0\rangle$ state might be incorrectly measured as $|1\rangle$, and vice versa. Readout errors are particularly detrimental in quantum algorithms that rely heavily on accurate measurement outcomes for decision-making and result interpretation.

The probability of a readout error can be characterized by a confusion matrix $M$, which quantifies the likelihood of each possible measurement outcome given the true state:

$$M = \begin{pmatrix} P(0|0) & P(1|0) \\ P(0|1) & P(1|1) \end{pmatrix},$$

where $P(a|b)$ denotes the probability of measuring outcome $a$ given the qubit was in state $|b\rangle$.

## A11 RELATED WORKS

### A11.1 QUANTUM ERROR MITIGATION

Traditionally, Zero-Noise Extrapolation (ZNE) is a widely implemented mitigation scheme. The essence of ZNE lies in deliberately introducing varying levels of noise into the quantum circuits and performing multiple executions of these circuits. Based on these noisy outcomes, ZNE employs extrapolation namely fitting techniques to estimate the ideal quantum state probability distribution in the absence of noise, whereby the Richardson extrapolation is a basic fitting method. However, ZNE requires high computational resources by necessitating multiple executions of quantum operations at varying noise levels, leading to higher resource consumption. It also heavily relies on the accuracy of the assumed noise model for extrapolation.

Czarnik et al. (2021) proposes Clifford Data Regression (CDR) to predict the noise-free observable. The majority of non-Clifford gates in the target circuit are (approximately) replaced with nearby Clifford gates, such that classically simulable training data can be efficiently constructed, as the Clifford gates are more friendly for efficient simulation. Then a linear regression model is employed that regresses the noisy expectation values of those circuits onto the ideal ones.

Kim et al. (2020) employs an ordinary Neural Network (NN) and a concatenated NN to learn the amount of error in the result measurement of quantum output states. However, the inputs of their NNs only consider the number of gates and ideal and noisy measurement information. Liao et al. (2024) benchmarks four models: linear regression, random forests, multi-layer perceptrons, and graph neural networks, to minimize the sum squared error between the mitigated and ideal expectation values. They convert the transpiled circuit into an acyclic graph and take it as input to GNNs. However, they struggle to handle large-scale quantum circuits effectively because they train the model by mimicking other scalable QEM methods, such as ZNE, which limits their model from outperforming the mimicked method. In contrast, our approach ensures scalability and leverages inverse circuits to handle large-scale problems effectively.

### A11.2 GRAPH TRANSFORMER

In recent years, Graph Transformers have emerged as a powerful tool for processing graph-structured data by combining the strengths of both Graph Neural Networks (GNNs) and Transformer architectures. This hybrid model has demonstrated strong performance across a wide range of tasks, including node classification, graph generation, and molecular property prediction. Graph Transformers can be classified into message-passing-based and non-message-passing-based models, based on their approach to information propagation in graphs.

Message-passing-based models integrate traditional GNN frameworks, where each node aggregates information from its neighbors. GAT (Veličković et al., 2018) is an early example, using attention mechanisms to assign weights to neighboring nodes during aggregation. Graphormer (Ying et al., 2021) extends this by adding global attention to enhance performance in tasks like molecular property prediction. GPS (Rampášek et al., 2022), another variant, merges message-passing with self-attention

for efficient local and global interaction handling. Liao & Smidt (2022) proposes an equivariant graph attention transformer for 3D structures, which can be applied to tasks related to molecules and proteins (Yang et al., 2022a; Wu et al., 2024a).

However, graph transformers that use message-passing inherit known issues of message-passing, such as over-smoothing, over-squashing, and expressive power limitations, and differ significantly from Transformers used in other domains, thus making a transfer of research advances more difficult. In contrast, non-message-passing models rely solely on the Transformer's self-attention mechanism, bypassing iterative neighbor aggregation. GraphiT (Mialon et al., 2021), for instance, employs self-attention and positional encodings without traditional message-passing. Graph-BERT (Zhang et al., 2020) applies Transformer-style attention directly to node embeddings, capturing long-range dependencies more effectively. GTNcite (Yun et al., 2019) focuses on hierarchical graph representation using edge-wise attention, optimizing computational efficiency. For non-message-passing non-message-passing, learning meaningful attention scores typically requires capturing important positional or structural relationships between nodes, and strong positional encodings are challenging to design since the structure and symmetries of graph data are fundamentally different from that of other (Euclidean) domains. Nevertheless, in the field of quantum noise mitigation, we can not only model quantum circuits as graphs to incorporate structural information but also generate effective positional encodings for each node based on the execution order of the quantum circuit. The proposed approach effectively addresses the challenge.

### A11.3 Graph Neural Networks for Electronic Design Automation

Graph Neural Networks (GNNs) are specialized models designed to process graph-structured data through trainable parameters, which have served widespread graph applications like combinatorial optimization (Li et al., 2024; 2025; 2023) and electronic design automation (Xie et al., 2021; Cheng et al., 2022; Guo et al., 2022; Wang et al., 2022a). As GNN research progresses, novel architectures are being introduced that target graph topological properties (Ye et al., 2023a;b; Zhao et al., 2023) and scalability to large graphs (Wu et al., 2022; 2023a;b). Given the inherent (hyper-)graph structure of circuit netlists, GNNs are naturally suited for applications in Electronic Design Automation (EDA), particularly in the context of Power, Performance, and Area (PPA) estimation. Early GNN models, such as GCN (Defferrard et al., 2016; Kipf & Welling, 2017), extend the concept of convolutions from low-dimensional regular grids to high-dimensional irregular domains, while GAT (Veličković et al., 2018) introduces attention mechanisms to learn adaptive weights for neighboring nodes. GCNII (Chen et al., 2020) addresses the over-smoothing issue through initial residual connections and identity mapping, whereas GINE (Hu et al., 2020), an extension of GIN (Xu et al., 2018), incorporates edge features to enhance expressiveness and was originally designed to tackle graph isomorphism challenges. Despite their success, these models face limitations when applied to circuit graphs with long timing paths due to restricted local receptive fields, which could lead to over-smoothing problems.

To overcome these limitations, recent works have developed GNN variants specifically tailored for EDA tasks. Net$^2$ (Xie et al., 2021) leverages graph attention networks to estimate net length during the pre-placement phase. CircuitGNN (Yang et al., 2022b) represents circuits as heterogeneous graphs and introduces a topo-geom message-passing paradigm to predict wire length and congestion. Timing-GCN (Guo et al., 2022) models a GNN inspired by timing engines, enabling effective slack prediction in pre-routing stages. LHNN (Wang et al., 2022a) employs a lattice hyper-graph neural network for routing congestion prediction. Finally, GTN (Fan et al., 2024) proposes a Graph-Transformer-based architecture as a surrogate model for circuit evaluation, achieving significant acceleration in circuit topology optimization without compromising performance.

## A12 Experiment Settings

### A12.1 setups

The quantum circuits in our experiments are executed on IBM's backend. Our graph-learning QEM algorithm is implemented on a system running Ubuntu 16.04, with CUDA 11.0, PyTorch 1.13.0, and PyTorch Geometric 2.3.1. Most experiments are conducted on an NVIDIA 4090 GPUs with 24GB of memory.

### A12.2 METRICS CALCULATION

To evaluate the performance of our proposed model, we employ two key metrics: Mean Absolute Error (MAE) and Circuit Improvement Ratio (CIR). These metrics provide a comprehensive assessment of the model's effectiveness in mitigating errors within quantum circuits.

#### A12.2.1 MEAN ABSOLUTE ERROR (MAE)

The Mean Absolute Error (MAE) quantifies the average magnitude of errors between the predicted values and the actual ground truth values, without considering their direction. It is defined mathematically as:

$$\text{MAE} = \frac{1}{N} \sum_{i=1}^{N} |\hat{y}_i - y_i| \tag{13}$$

where $N$ is the total number of quantum circuits evaluated. $\hat{y}_i$ represents the predicted value for the $i$-th circuit. $y_i$ denotes the actual ground truth value for the $i$-th circuit.

A lower MAE indicates higher prediction accuracy, reflecting the model's ability to closely approximate the true values of the quantum circuits.

#### A12.2.2 CIRCUIT IMPROVEMENT RATIO (CIR)

The Circuit Improvement Ratio (CIR) measures the proportion of quantum circuits that have been successfully improved through the error mitigation process. It is calculated using the following formula:

$$\text{CIR} = \frac{\text{Number of Improved Circuits}}{\text{Total Number of Circuits}} \times 100\% \tag{14}$$

where **Number of Improved Circuits** refers to the count of circuits that exhibit enhanced performance post-error mitigation compared to their original state. **Total Number of Circuits** is the overall number of circuits subjected to error mitigation.

A higher CIR signifies a more effective error mitigation strategy, indicating that a greater percentage of circuits have benefited from the proposed enhancements.

### A12.3 DATA PREPARATION

We provide detailed descriptions of the experimental settings used in our work to facilitate future reproduction. We cover the circuit datasets, noise models, experimental configurations across different quantum system scales, and the hyperparameters used in our method.

#### A12.3.1 CIRCUITS CONSTRUCTION

**Trotterized Circuits Construction.** We focus on the Trotterized dynamics of the one-dimensional transverse-field Ising model (TFIM) as described in Chakrabarti et al. (1996). The Trotterized circuits used in our experiments simulate the dynamics of a spin chain governed by the Hamiltonian:

$$\hat{H} = -J \sum_j \hat{Z}_j \hat{Z}_{j+1} + h \sum_j \hat{X}_j = -J\hat{H}_{ZZ} + h\hat{H}_X,$$

where $J$ represents the exchange coupling between neighboring spins, and $h$ denotes the transverse magnetic field. We generated multiple circuit instances by randomly assigning the parameters $J$, $h$, and the evolution time $t$, with coupling strengths uniformly sampled from the paramagnetic phase ($J < h$). The number of Trotter steps varied within specified ranges, determining the depth of the circuits.

**Random Circuits Construction** We utilize the random-structure circuits to demonstrate the generality of GTranQEM. In this setting, circuits in a dataset consist of at most $N_q$ qubits and most $N_{gate}$. The allowed gates are chosen from {Pauli-X, Pauli-Y, Pauli-Z, Hadamard, RX, RY, RZ, Controlled-NOT, Controlled-Z, SWAP, RZZ}. The types of, the orders of, and the qubits being operated by quantum gates are randomized within each circuit. Rotation angle for rotation gates are randomly selected from $[0, 2\pi]$

A12.3.2    CIRCUIT DATASET CONSTRUCTION

We constructed various circuit datasets to evaluate the performance of GTranQEM across different system sizes and experimental settings. The datasets are categorized based on the circuit types and scales:

**Max-6-Random and All-6-Trotter Datasets**    For the *max-6-random* and *all-6-trotter* datasets, we generated circuits with up to 6 qubits. The training set consists of 750 circuits with a maximum length of 150 gates, randomly selected from the allowed gate list. We used an additional 150 circuits for validation and 300 circuits for testing. These datasets are used to evaluate the performance of GTranQEM on small-scale quantum systems.

**All-15-Trotter Dataset**    The *all-15-trotter* dataset comprises circuits with 15 qubits. We employed 260 circuits for training, 40 circuits for validation, and 100 circuits for testing. This dataset allows us to assess the scalability of GTranQEM to medium-scale quantum systems.

**All-50-Trotter Dataset**    For large-scale systems, we constructed the *all-50-trotter* dataset, consisting of 50-qubit circuits. We created 20 Trotter circuits with Trotter steps ranging from 1 to 3. Each circuit was initialized with 10 random initial states, resulting in 200 training circuits. Two original circuits were used for validation, and the remaining 18 circuits were used for testing. This dataset was executed on a real quantum device (`IBM_Brisbane`) to evaluate the practical applicability of GTranQEM.

**Circuit-Scale Extrapolation Dataset**    We constructed a dataset with circuits having Trotter steps ranging from 1 to 20. We used 260 circuits with Trotter steps from 1 to 15 for training, 40 circuits for validation, and 100 circuits with Trotter steps from 16 to 20 for testing. This setting allows us to evaluate the ability of GTranQEM to extrapolate to deeper circuits beyond those seen during training.

**Noise Scale Amplification Dataset**    We applied noise scaling factors $\Lambda = \{1, 2, 3\}$ to augment the training set without generating additional circuits. We used 200 circuits for training, resulting in 600 data points after noise scaling, and 200 circuits for testing. This dataset allows us to assess the robustness of GTranQEM to varying noise levels.

**6-Qubit No-Ideal Setting Dataset**    For the 6-qubit no-ideal setting, we constructed a training dataset of 400 inverse-composed circuits, each with one random initial state. We used 100 circuits for validation and 300 circuits for testing. This dataset was used to evaluate the performance of GTranQEM when ideal expectation values are not available.

**Remarks.** In all experimental settings, we ensured that there was no overlap between the training, validation, and testing datasets. This separation is crucial for obtaining unbiased evaluations of the model's performance.

A12.3.3    NOISE SETTINGS

We evaluated the performance of GTranQEM under various noise settings to best testify the performance of GTranQEM. The noise models used in our experiments are:

1) **Fake Provider Setting** `FakeHanoiV2` is used, which emulates the noise characteristics of the IBM Hanoi quantum device.

2) **Incoherent error-only setting** Incoherent noise was simulated by applying a combination of Pauli X and depolarizing errors on all single-qubit and two-qubit gates. The error rates followed the statistics of the Sycamore quantum device (Arute et al., 2019), as listed in Table 3.

3) **Mixed Noise Setting** Mixed noise combines incoherent noise with additional error sources to simulate a more realistic noise environment.

4) **Coherent Noise Setting** Coherent noise was introduced by adding systematic over-rotation errors to the double-qubit gates (CX, CY, CZ, and Swap gates) with an average over-rotation angle of $0.02\pi$ with a combination of the noise model extracted from the Fake Lima provider.

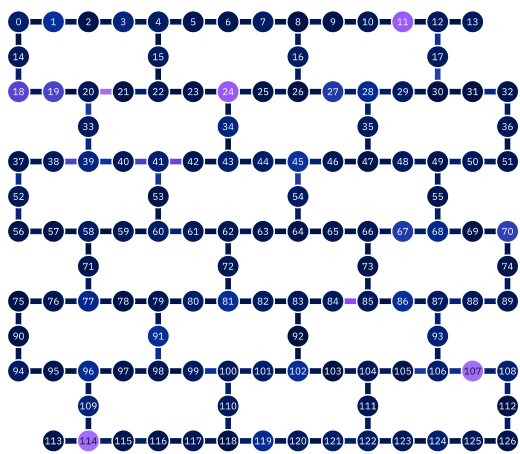

Figure 5: The topology of qubits of IBM_Brisbane.

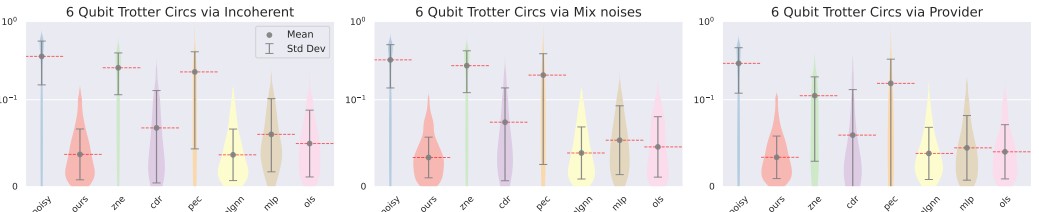

Figure 6: Visualization of the distribution of AEs for all-6-trotter circuits under three noise settings. Distributions closer to zero represent better performance.

Table 3: Error rates for Incoherent and Mixed noise settings based on the Sycamore quantum device Arute et al. (2019).

| System | # of Qubits | Single-Qubit Gate Error | Two-Qubit Gate Error | Readout Error |
|---|---|---|---|---|
| Sycamore Arute et al. (2019) | 53 | 0.16% | 0.62% | 3.8% |

5) **IBM's Brisbane** For experiments on real quantum devices, we executed circuits on IBM_Brisbane, a 127-qubit quantum computer provided by IBM. The details of the quantum device are as follows:

1. **Topology**: IBM_Brisbane has a 127-qubit architecture. We utilized the Qiskit quantum cloud platform, which optimizes the 50-qubit circuits and maps them onto the 127 qubits using the generate_preset_pass_manager function. The topology of these qubits is illustrated in Fig. 5.

2. **Coherence Times**: The median relaxation time $T_1$ is 213.21 $\mu$s, and the median dephasing time $T_2$ is 147.61 $\mu$s, indicating relatively long coherence times suitable for executing deep quantum circuits.

3. **Fidelities**: IBM_Brisbane provides the metric *2Q Error (layered)*, representing the average two-qubit error rate per layered gate (EPLG) for a 100-qubit chain. The 2Q Error (layered) for IBM_Brisbane is $2.06 \times 10^{-2}$.

4. **Additional Parameters**: The median echoed cross-resonance (ECR) error is $8.340 \times 10^{-3}$, the median single-qubit gate (SX) error is $2.345 \times 10^{-4}$, and the median readout error is $1.370 \times 10^{-2}$.

### A12.4   EXPERIMENTAL CONFIGURATIONS

We conducted experiments across three scales of quantum systems—small-scale (4–6 qubits), medium-scale (15 qubits), and large-scale (50 qubits)—under both standard and non-standard settings.

#### A12.4.1   STANDARD SETTINGS

We evaluated GTranQEM under three standard settings—small-scale, medium-scale, and large-scale systems—to demonstrate its effectiveness across different scales. For small-scale systems, we used datasets like *max-6-random*, *all-6-trotter*, and *all-4-trotter* under various noise conditions, including *fake provider*, *incoherent noise only*, *mixed noise*, and *coherent noise*. In medium-scale systems, we assessed the scalability of GTranQEM using the *all-15-trotter* dataset under *incoherent noise only*. For large-scale systems, we validated GTranQEM on real quantum devices by performing error mitigation on the *all-50-trotter* dataset executed on `IBM_Brisbane`.

#### A12.4.2   NON-STANDARD EXPERIMENTAL SETTINGS

To further demonstrate the generality and robustness of GTranQEM, we conducted experiments under non-standard settings.

**Circuit Depth Generality**   We testify the generality of GTranQEM by performing mitigation tasks over the circuit-scale extrapolation datasets. Training was conducted on circuits with Trotter steps from 1 to 15, and testing was performed on circuits with Trotter steps from 16 to 20. The results demonstrated in Tab. 2 and Fig. 4 (c), demonstrate GTranQEM's capability of generalizing to deep circuits.

**Evaluation Without Ideal Expectation Values**   We assessed the performance of GTranQEM in the absence of ideal expectation values (EVs) by employing the *6-qubit no-ideal setting dataset* to evaluate the effectiveness of our proposed circuit inverse composition scheme.

Initially, we conducted experiments on the 6-qubit dataset without ideal EVs to provide insights into error mitigation for larger quantum circuits. The results, presented in Fig.7, indicate that GTranQEM performs effectively under these conditions. Notably, GTranQEM achieves a superior performance over the traditional QEM methods showing the effectiveness of our circuit inverse composition paradigm; GTranQEM outperforms other machine learning-based models that operate under the same paradigm in this setting, highlighting the efficacy of our model architecture. It is important to mention that the 6-qubit setting serves primarily to validate the proposed paradigm, as ideal EVs are readily obtainable for such small quantum systems. Indeed, GTranQEM demonstrates strong performance when trained with ideal EVs as labels (see Figs.3 and 6).

When extending the mitigation to larger qubit systems, the advantages of our paradigm become more pronounced. As depicted in Fig. 4,(f), GTranQEM surpasses Zero-Noise Extrapolation (ZNE), which is among the few traditional QEM methods feasible within reasonable execution times for large-qubit circuits with general structures. Although multilayer perceptron (MLP) and graph neural network (GNN) models can also be utilized under our paradigm for larger systems, they exhibit inferior performance compared to GTranQEM, underscoring the effectiveness of our model design.

#### A12.4.3   HYPERPARAMETERS

The hyperparameters of the GTranQEM model layers for each experimental setting are provided in Table 4. These parameters were chosen based on preliminary experiments to optimize performance for each specific setting.

## A13   TIME ANALYSIS

Comparing the computational time between machine learning (ML)-based and classical quantum error mitigation (QEM) methods is challenging due to inherent differences in their workflows.

Classical QEM approaches typically handle quantum circuits individually, potentially involving extensive mitigation time due to circuit-specific tuning, multiple circuit executions, or additional

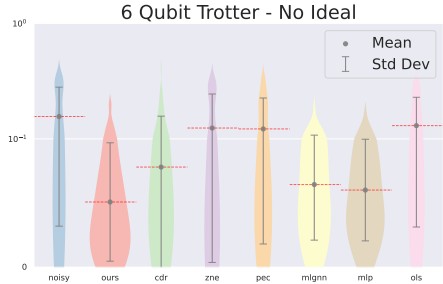

Figure 7: Distributions of AEs for no ideal EV setting of all-6-trotter circuits under the mixed noise setting.

Table 4: Hyperparameters of GTranQEM layers for each experimental setting.

| Experiment Name | Number of Layers | Hidden Dimension |
|---|---|---|
| All-6-Trotter, Mixed Noise | 1 | 24 |
| All-6-Trotter, Incoherent Noise | 1 | 128 |
| All-6-Trotter, Provider Noise | 1 | 32 |
| Max-6-Random, Mixed Noise | 1 | 64 |
| Max-6-Random, Incoherent Noise | 2 | 64 |
| Max-6-Random, Provider Noise | 2 | 64 |
| All-15-Trotter, Incoherent Noise | 2 | 256 |
| Circuit-Scale Extrapolation | 1 | 64 |
| Noise Amplification | 1 | 64 |
| 6-Qubit, No Ideal Expectation Values | 1 | 64 |
| 50-Qubit, No Ideal Expectation Values | 2 | 512 |

Table 5: Training and inference times for different methods over various datasets. The times for ZNE and CDR correspond to the mitigation times applied to the same number of circuits during inference. All quantum circuit simulations are executed via IBM Qiskit. GTraQEM is implemented on a computing system running Ubuntu 18.04.6 with CUDA version 12.1. The majority of experiments are performed on an NVIDIA GeForce RTX 4090 GPU with 24 GB of memory.

| Method | 4-qubit, 300 circuits | | 6-qubit, 300 circuits | | 15-qubit, 100 circuits | |
|---|---|---|---|---|---|---|
| | Training | Inference | Training | Inference | Training | Inference |
| GTraQEM | 21.90 s | 0.1760 s | 38.70 s | 0.1868 s | 18.64 s | 0.121 s |
| OLS | 3.42 s | 0.0734 s | 5.53 s | 0.0864 s | 2.21 s | 0.022 s |
| ML-GNN | 31.57 s | 0.1473 s | 43.54 s | 0.1916 s | 26.41 s | 0.104 s |
| MLP | 25.57 s | 0.0896 s | 32.71 s | 0.1037 s | 15.94 s | 0.098 s |
| ZNE | n/a | 548.78 s | n/a | 782.03 s | n/a | 2166.46 s |
| CDR | n/a | 1841.34 s | n/a | 1796.65 s | n/a | 17405.38 s |

qubit overheads. Moreover, their performance is often significantly influenced by the number of qubits and the scale of the circuits.

In contrast, ML-based approaches consist of two primary stages: model training and inference. Once the model is trained, inference (i.e., the actual error mitigation) can be performed quickly. For instance, after fine-tuning GTraQEM for 6-qubit Trotter circuits executed under the Fake Hanoi Provider, we can mitigate the results of new 6-qubit Trotter circuits on that device in negligible time. Consequently, a direct and fair comparison between these two types of methods is not straightforward.

To provide a comparative perspective, we present in Table 5 the training and inference times required by each ML-based method, alongside the mitigation times for Zero-Noise Extrapolation (ZNE) and Clifford Data Regression (CDR) when applied to the same number of circuits during inference. The table demonstrates that GTraQEM achieves computational efficiency comparable to other ML-based

| Index | PE | Edge Feature | QCR node | MP | MAE | Std | CIR |
|-------|----|--------------|----------|----|--------|--------|--------|
| 0 | | | | | 0.0416 | 0.0347 | 0.9033 |
| 1 | | | | ✓ | 0.0440 | 0.0283 | 0.8867 |
| 2 | | | ✓ | | 0.0547 | 0.0467 | 0.9033 |
| 3 | | | ✓ | ✓ | 0.0332 | 0.0250 | 0.9100 |
| 4 | | ✓ | | | 0.0425 | 0.0364 | 0.9233 |
| 5 | | ✓ | | ✓ | 0.0434 | 0.0279 | 0.8433 |
| 6 | | ✓ | ✓ | | 0.0547 | 0.0467 | 0.9033 |
| 7 | | ✓ | ✓ | ✓ | 0.0324 | 0.0261 | 0.8967 |
| 8 | ✓ | | | | 0.0360 | 0.0282 | 0.9267 |
| 9 | ✓ | | | ✓ | 0.0448 | 0.0328 | 0.8867 |
| 10 | ✓ | | ✓ | | 0.0359 | 0.0305 | 0.9200 |
| 11 | ✓ | | ✓ | ✓ | 0.0373 | 0.0305 | 0.8967 |
| 12 | ✓ | ✓ | | | 0.0380 | 0.0286 | 0.9233 |
| 13 | ✓ | ✓ | | ✓ | 0.0323 | 0.0277 | 0.9133 |
| 14 | ✓ | ✓ | ✓ | | 0.0350 | 0.0314 | 0.9133 |
| 15 | ✓ | ✓ | ✓ | ✓ | **0.0315** | **0.0272** | **0.9267** |

Table 6: Ablation study results showing the effect of each module on the QEM performance. The check marks indicate the inclusion of the respective module.

methods while delivering superior mitigation performance. While GTraQEM's inference time is marginally slower than that of the MLP and ML-GNN models, all three methods achieve inference times within one second, rendering the differences practically insignificant.

## A14 ABLATION STUDIES

We modify the classical graph transformer architecture with quantum-native modules to construct a graph transformer specialized for QEM tasks. To assess the impact of each proposed module, we conduct comprehensive ablation studies focusing on the quantum-native positional encoding (PE), qubit-wise edge features, the Quantum Circuit Representation (QCR) node, and the message passing (MP) scheme.

For the experiments, we utilize 6-qubit Trotter circuits subjected to mixed noise damage to perform the error mitigation task. As a baseline, we employ the Message Passing Neural Network (MPNN) Gilmer et al. (2017) as a standard graph transformer model without any quantum adaptive modules for comparison. We systematically add or remove each module to observe its influence on performance.

When the QCR node is turned off, we replace it with a standard graph mean pooling layer to aggregate node features. Additionally, when the message passing scheme is disabled, we use the standard adjacency matrix instead of the structure matrix to guide the training process.

Tab. 6 summarizes the results of our ablation studies. Overall, the inclusion of quantum-native modules significantly enhances the performance of our model, GTranQEM, on QEM tasks. Specifically, GTranQEM achieves a Mean Absolute Error (MAE) of **0.0315 ± 0.0272**, noticeably outperforming the baseline MPNN, which attains an MAE of $0.0416 \pm 0.0347$. We also report the highest Circuit Improvement Ratio (CIR), demonstrating the effectiveness of our approach.

