# OpenReview forum: "Beyond Circuit Connections: A Non-Message Passing Graph Transformer Approach for Quantum Error Mitigation"
_ICLR.cc/2025/Conference — ICLR 2025 Poster_

### Official Review · Reviewer_KKBh · 2024-10-30

**Soundness:** 3
**Presentation:** 2
**Contribution:** 3
**Rating:** 6
**Confidence:** 3

**Summary:**

This work presents a learning-based quantum error mitigation method by leveraging quantum-specific positional encoding and a structure information matrix for transformer to suppress the noise when estimate the expectation value.

**Strengths:**

this work exploits circuit topology in learning-based QEM empowered by multi-head attention and provides comprehensive experiments and shows the out-performance.

**Weaknesses:**

1. since it does not provide the related codes, the reproducibility is unknown.
2. while this work applies the various exist techniques to enhance the QEM which may be interesting for the community of quantum computation, it does not propose some novel structure or interesting theoretical findings which have widely applications.

**Questions:**

1. what does Fig.2. want to express? It is unclear the inverse quantum circuit used in here.
2. In Fig1. does the model use DAG with learnable node(QCR Node) to construct parametrized structure matrix? does the output of GTraQEM module concatenates with $x_{QCR}^L$ to form the input of the regression module? and how to construct $x_{QCR}^{L}$ ?
3. what's the sample complexity of such learning-base QEM model?

---

> ### Author Response · Authors · 2024-11-20
> **Responses to Reviewer KKBh (1/3)**
>
> Thank you for your valuable comments. We try to answer the questions one by one and hope we can alleviate your concerns.
>
> ### Responses to the Questions
>
> ***> Q1: "What does Fig. 2 want to express? It is unclear the inverse quantum circuit used in here."***
>
> We apologize for the misleading layout, as **Fig. 2 helps illustrate the data augmentation techniques we mentioned in Section 4.1.1 of the manuscript**. Namely, it visualizes how to append the inverse of a quantum circuit to the original circuit to construct training circuits when ideal EVs of circuits are hardly obtained (usually when the qubit number of the circuits is large).
>
> To be specific, ML-based QEM has the model training and inference stages, and **the labels (ideal EVs) of the training circuits are required for supervision** to guarantee the model's performance during the inference (i.e., the mitigation) stage. However, obtaining ideal EVs as labels can be difficult when the qubit number of circuits is large and ML models usually require more data for training. Accordingly, such a lack of training labels accordingly limited the application of ML-based QEM models.
>
> In this manuscript, by employing circuit inverse composition, we enable **label-efficient training**, allowing the acquisition of ideal EVs of training circuits without execution, which greatly reduces the difficulty in obtaining labels as the scale of the circuit increases.
>
> In summary, this approach extends the applicability of ML-based QEM to larger qubit systems while maintaining performance via label-efficient training, as demonstrated with our 50-qubit experiments on IBM Brisbane.
>
> For a more detailed explanation how the inverse quantum circuit and its role in our method, please refer to Section 4.1.1 of the manuscript. **Our work is the first, to our knowledge, to use this method to reduce dependence on ideal EVs in training ML-based QEM methods.**
>
>
> ***> Q2: "In Fig1. does the model use DAG with learnable node(QCR Node) to construct parametrized structure matrix? does the output of GTraQEM module concatenates with to form the input of the regression module? and how to construct $x_{QCR}^{L}$."***
>
> Fig. 1 uses the DAG with a learnable node (QCR node). In other words, the QCR node can be considered as a node in the input graph, but with the difference that its features are learnable. Initially, the features of the QCR node are randomly initialized, while the features of the other nodes are predefined. After passing through $L$ layers of our graph Transformer, the features of each node are updated, and at the same time, the QCR node's features are also updated, resulting in $x_{QCR}^{L}$. Then, $z = [x_{QCR}^{L} || y_{noisy}]$ is fed into a regression module consisting of two fully connected layers with a nonlinear activation function in between.
>
> ***> Q3:"what's the sample complexity of such a learning-base QEM model?"***
>
> We apologize for not being sure about what **"sample complexity"** you are referring to. We here discuss the time complexity of all QEM methods we utilized in this manuscript. If we misinterpret your question, please let us know and we are happy to add further clarification.
>
> #### **Time complexity**
>
> Comparing the computational time between ML-based and classical QEM methods poses challenges due to **inherent differences in their workflows**.
>
> - Classical QEM approaches handle circuits **one at a time**, perhaps involving extensive mitigation time due to circuit-individual tuning, multiple circuit executions, or extra qubit overheads. Further, most of them are easily influenced by the number of qubits and the scale of circuits.
> - ML-based approaches involve two stages, **model training** and **inference**. **Once the model is well-trained, the inference (i.e., the actual mitigation) occurs extremely quickly.** For example, after fine-tuning GTraQEM for 6-qubit Trotter circuits executed under the Fake Hanoi Provider, we can mitigate the results of new 6-qubit Trotter circuits on that device in negligible time. Accordingly, a direct and fair comparison of these two types of methods is not straightforward.
>
> We now provide a table showing the training and inference times required by each ML-based method, alongside the mitigation times for ZNE and CDR applied to the same number of circuits during inference for comparison. **The table demonstrates that GTraQEM achieves efficiency comparable to other ML-based methods while showing stronger mitigation performance.** While GTraQEM’s inference is marginally slower than MLP and MLGNN, all three methods achieve inference times within one second, rendering the difference insignificant.

---

> > ### Comment · Reviewer_KKBh · 2024-11-25
> >
> > Thank you for the response and mostly address my concern. For Q3, the sample complexity in my question corresponds to the scale of the training size.

---

> > > ### Author Response · Authors · 2024-11-25
> > > **Discussion about the scale of the training size**
> > >
> > > Thank you for the clarification and your insightful question.
> > >
> > > ### **Training Dataset Statistics**
> > >
> > > We employed several datasets to train our model, with their statistics summarized in the table below:
> > >
> > > | Dataset          | Number of Graphs (Circuits) | Average Number of Nodes per Graph | Node Feature Dimension |
> > > | ---------------- | --------------------------- | --------------------------------- | ---------------------- |
> > > | All-4-Trotter    | 750                         | 145                               | 18+18                     |
> > > | Max-6-Random     | 750                         | 89                                | 20+20                     |
> > > | All-6-Trotter    | 750                         | 232                               | 20+20                     |
> > > | All-15-Trotter   | 260                         | 316                               | 29+29                     |
> > > | All-50-Trotter   | 180                         | 938                               | 64+64                    |
> > >
> > > *Note: Validation datasets are not included in this table.*
> > >
> > > The number of graphs (circuits) in each dataset can be manually adjusted. The graph size and node feature of each graph are determined by **the original circuit** and **the encoding scheme** described in **Section 3.2** of our manuscript. Specifically:
> > >
> > > - **Number of Nodes per Graph**:
> > >   $$
> > >   \text{Node Count} = 2 \times \text{Number of Qubits} + \text{Number of Gates} + 1.
> > >   $$
> > >   The additional 1 accounts for the QCR node.
> > >
> > >
> > > - **Node Feature Dimension**:
> > >   $$
> > >   \text{Raw Feature Dimension} = \text{Number of Allowed Gate Types} + \text{Number of Qubits} + 1.
> > >   $$
> > >   The additional 1 accounts for the rotation angle if exists. The positional encoding in the same dimension as the raw feature is concatenated to the raw feature to form the final node feature for the model.
> > >
> > > ### **Computational Resources**
> > >
> > > Each experiment could be conducted using one NVIDIA GeForce RTX 4090 GPU with 24 GB of memory.
> > >
> > > ### **Impact of Training Size on Mitigation Performance**
> > >
> > > To evaluate how the size of the training dataset influences the mitigation performance, we conducted an analysis using the **All-6-Trotter** dataset under the Fake Provider noise setting. We varied the number of training circuits and measured the mean absolute error (MAE) of the mitigated results among the test sets. The relationship between the training data size and the mitigation performance is illustrated in [this figure](https://postimg.cc/BjcvV8RK).
> > >
> > > Our findings are as follows:
> > >
> > > - **Performance Improvement with Increased Data**: There is a visible improvement in mitigation performance as the number of training circuits increases.
> > > - **Diminishing Returns Beyond a Threshold**: The impact of adding more training data becomes minimal once the number of training circuits becomes "enough" (approximately 700 in this case). Beyond this point, the performance gains plateau, indicating that the model has sufficiently learned.
> > >
> > >
> > > We hope this explanation adequately addresses your question regarding the scale of training size. Please let us know if you have any further inquiries and we would be happy to discuss further.

---

> > > > ### Comment · Reviewer_KKBh · 2024-11-26
> > > >
> > > > Thank you for the reply and I would like to raise my score.

---

> > > > > ### Author Response · Authors · 2024-11-26
> > > > >
> > > > > Dear Reviewer KKBh,
> > > > >
> > > > > We would like to express our sincere gratitude for your support of our work. Your constructive suggestions have been invaluable in refining our manuscript, and we greatly appreciate the time and effort you dedicated to reviewing it.
> > > > >
> > > > > Thank you once again for your support.
> > > > >
> > > > > Best wishes,
> > > > >
> > > > > The Authors

---

> ### Author Response · Authors · 2024-11-20
> **Responses to Reviewer KKBh (2/3)**
>
> |             | 4q - 300circs |               | 6q - 300circs |               | 15q - 100circs |               |
> | ----------- | ------------- | ------------- | ------------- | ------------- | -------------- | ------------- |
> | **Method**  | **Training**  | **Inference** | **Training**  | **Inference** | **Training**   | **Inference** |
> | **GTraQEM** | 21.90s        | 0.1760s       | 38.70s        | 0.1868s       | 18.64s         | 0.121s        |
> | **OLS**     | 3.42s         | 0.0734s       | 5.53s         | 0.0864s       | 2.21s          | 0.022s        |
> | **ML-GNN**  | 31.57s        | 0.1473s       | 43.54s        | 0.1916s       | 26.41s         | 0.104s        |
> | **MLP**     | 25.57s        | 0.0896s       | 32.71s        | 0.1037s       | 15.94s          | 0.098s        |
> | **ZNE**     | n/a           | 548.78s       | n/a           | 782.03s       | n/a            | 2166.46s     |
> | **CDR**     | n/a           | 1841.34s      | n/a           | 1796.65s      | n/a            | 17405.38s     |
>
> *Note: All quantum circuit simulations are executed via IBM qiskit. GTraQEM is implemented on a computing system running Ubuntu 18.04.6, equipped with CUDA version 12.1 The majority of experiments are performed on an NVIDIA GeForce RTX 4090 GPU, which has 24 GB of memory.*
>
> We hope this additional information clarifies the computational efficiency of our proposed method compared to other error mitigation techniques and will add the analysis of the time complexity to our manuscript.
>
> ### Responses to the Weaknesses
>
> ***> W1: "Since it does not provide the related codes, the reproducibility is unknown."***
>
> Please check the reply, "Repo of GTraQEM" below.

---

> ### Author Response · Authors · 2024-11-20
> **Responses to Reviewer KKBh (3/3)**
>
> ***>W2: While this work applies the various exist techniques to enhance the QEM which may be interesting for the community of quantum computation, it does not propose some novel structure or interesting theoretical findings which have widely applications.***
>
> We would like to emphasize the key innovation of our paper lies in the targeted design of the model for the characteristics of quantum, including the following aspects:
>
> 1. **Novel Encoding Scheme**: We introduce a unique encoding approach where quantum circuits are represented as directed acyclic graphs (DAGs), with qubit-wise measurement outcomes incorporated as edge features. This encoding captures both the circuit topology and the measurement information in a unified framework, enhancing the model's ability to learn complex quantum system behaviors.
>
> 2. **Customized Application of Non-Message Passing Graph Transformer**: We are among the first to apply a non-message passing graph transformer architecture to the QEM problem. Traditional message-passing mechanisms focus on the neighborhood without capturing long-range dependencies inherent in quantum circuits. In contrast, GTraQEM leverages **both the circuit's topology** via the graph encoding and the **long-range dependencies** via the non-message passing scheme. We further include a learnable QCR node to facilitate graph-level feature aggregation, making GTraQEM particularly suitable for modeling quantum systems.
>
> 3. **Refinement of the Learning-Based QEM Paradigm**: Previous learning-based QEM methods [1] often rely on obtaining ideal expectation values through classical simulation, which becomes impractical for larger circuits due to the exponential scaling of computational resources. We propose to refine the paradigm by constructing the training set with standard circuits composed with their inverse. This approach allows us to obtain ideal EVs as labels of the training circuits for supervision without circuit executions, as the initial state becomes the final quantum state. This significantly reduces ML-based methods' reliance on the ideal EVs, achieving label-efficient training.
>
> 4. **Comprehensive ML Perspective and Effective Pipeline**: Given the limited number of existing ML-based QEM methods, our work provides a fresh perspective by thoroughly exploring the intersection of Graph learning and QEM. We propose an effective and complete pipeline for error mitigation across various circuit types and scales, highlighting its potential for wide applicability in the quantum computing community.
>
> Additionally, other reviewers have also mentioned the novelty of our approach. For example, Reviewer PHEy remarked, "The technique used in this paper is novel," and Reviewer xXV6 commented, "GTraQEM introduces a unique non-message-passing graph transformer approach and a circuit-to-graph encoding scheme for quantum error mitigation." These observations emphasize the innovative nature of our methodology.
> We shall add the novelty analysis to the updated pdf.
>
> **References**
>
> [1] Machine Learning for Practical Quantum Error Mitigation, Nat Mach Intell (2024).
>
>
> We appreciate your feedback and hope that this clarification highlights the innovative aspects and broad applications of our proposed method.

---

> ### Author Response · Authors · 2024-11-24
>
> Dear Reviewer KKBh,
>
> Thank you for your valuable feedback. As the discussion phase deadline nears, we look forward to your responses. If there are any additional questions or concerns, we would love to have a further discussion.
>
> Best wishes,
>
> the authors

---

> ### Author Response · Authors · 2024-11-25
>
> Dear Reviewer KKBh,
>
> Thank you for your dedicated review of our manuscript. We hope our reply concerning the training size scale has adequately addressed your inquiries.
>
>  With the discussion phase deadline fast approaching, we look forward to your feedback. If there are any remaining concerns or if further discussion is needed, please let us know.
>
> Best regards,
>
> The Authors

---

### Official Review · Reviewer_PHEy · 2024-10-31

**Soundness:** 3
**Presentation:** 3
**Contribution:** 4
**Rating:** 8
**Confidence:** 4

**Summary:**

This work presents a novel quantum error mitigation (QEM) method, GTraQEM, which uses a machine-learning approach. GTraQEM encodes the quantum circuit into a graph, then uses a non-message-passing graph transformer to construct a QEM mitigator. The effect is proved by 1) numerical simulations using a quantum circuit simulator; 2) IBM's quantum computer, and the authors compares GTraQEM with many other existing QEM approaches.

QEM is a popular way to handle the quantum error in near-term quantum computers which are subject to noise, before a practical quantum error correction can be realized.  ML-based QEM is of great importance in the NISQ era because of its practicality in the experiment.

**Strengths:**

1. The technique used in this paper is novel, and the results are good. Inputting the circuit structure into the graph transformer to construct ML-based QEM is intuitive, which is also proved to be effective via numerical evidence.
2. This paper uses a practical test case (transverse-field Ising model), which is considered to be one of the most possible applications of quantum computers in NISQ-era.

**Weaknesses:**

1. Unclear experiment settings.

The experiment settings used in this paper are not clearly stated in the experiment section. This highly weakens the paper's conclusion. Because QEM is an area that is mainly based on experimental/numerical evidence, it is vital to clarify the methods and the settings when the authors do not provide any codes.

In my opinion, the results shown in the figures and tables are not that convincing for me. For example, In Fig. 3, the results show that ZNE does not have any effect in all experiments. This clearly contradicts with the famous known results such as [Nature volume 618, pages500–505 (2023)], which also applies the transverse-field Ising model and trotterisation.

2. The paper writing needs to be strengthened. For example, the 15-qubit circuits experiment in Table 2 seem to be confusing. Why do we need this?

3. Figure 3 is unclear. I can hardly recognize the color.

4. There are many typos. For example, 6 qubit should be 6-qubit (in Fig. 3), and the
broken sentence in line 408.

5. The limitations of the proposed approach should be discussed, including the extra cost caused by the acquisition of the training set, and the cost of the training process.

**Questions:**

A few concerning points are listed as follows, and I hope the authors can clarify these before I change my mind about this paper's decision.

1. The original data/code generated by this paper should be provided to improve reproducibility. Also, there should be an explanation about why ZNE does not have any effect in Fig. 3 and Table 1.

2. The y-axis in Fig. 4 is the absolute error of EV. However, panel (f) implies there is some distribution over 2.0. In fact, it should be always in the range [0,2] (considering the observable should be within [-1,1]).

3. Whether the type of the circuit matter in GTraQEM? How is the generalizability of the method?

4. As this paper uses an IBM's quantum computer (Brisbane), it is recommended to articulate the hardware's physical parameters to improve the soundness, e.g. the topology (including the 50 used qubits), the coherence time, the single-qubit and two-qubit fidelities during the experiment.

---

> ### Author Response · Authors · 2024-11-20
> **Responses to Reviewer PHEy (2/3)**
>
> ####  **Impact on the Evaluation of GTraQEM**
> We would like to emphasize that, **ZNE is neither the strongest nor the only baseline we include.** Our study incorporates multiple QEM baselines, encompassing both classical and ML-based methods. Some of these baselines (e.g., CDR, ML-GNN) exhibit competitive performance, occasionally approaching ideal results. The consistent outperformance of these baselines across four different noise settings, combined with our method's efficiency, demonstrates the effectiveness of GTraQEM.
>
> We hope this detailed response addresses your concerns and enhances your confidence in our work.
>
> **Reference:**
>
> [1] Evidence for the utility of quantum computing before fault tolerance, Nature, 2023.
>
> [2] Machine Learning for Practical Quantum Error Mitigation,  Nat Mach Intell (2024)
>
> ***> Q2: "The y-axis in Fig. 4 is the absolute error of EV. However, panel (f) implies there is some distribution over 2.0. It should be always in the range [0,2] (considering the observable should be within [-1,1])."***
>
> Thank you for bringing this to our attention! Such misleading is caused by Matplotlib's built-in estimation of the violin plot. **All absolute error values are indeed within the [0, 2] range**, consistent with the expectation that observables lie within [-1, 1]. Yet, violin plots use KDE to estimate the probability density function of the data. KDE smooths the data distribution, which can naturally create tails extending beyond the minimum and maximum data points. To provide clarity, we have included an additional [line plot](https://postimg.cc/FddpfXsT). We shall set the limit to 2 of the violin plots for clarity and apologize for any confusion.
>
> ***> Q3: Whether the type of the circuit matter in GTraQEM? How is the generalizability of the method?***
>
> In short, GTraQEM's effectiveness is not dependent on the specific type or structure of quantum circuits. It demonstrates strong generalizability across various circuit architectures and scales. Specifically:
>
> 1. **Independence from Circuit Structure:**
>     - **Circuit-Agnostic Design:** GTraQEM is engineered to perform effectively across diverse circuit types without relying on any specific circuit architecture.
>     - **Performance on Varied Circuits:** It consistently achieves strong results on both random unstructured circuits (Max-N-Qubit dataset) and structured circuits (All-N-Qubit Trotter circuits dataset), as shown in Tab. 1, Fig. 3, and Fig. 4(a) of our manuscript.
>     - **Robustness:** This consistent performance across different circuit types underscores the general applicability and robustness of GTraQEM.
>
> 2. **Generalizability Across Circuit Scales:**
>     - **Circuit-Scale Extrapolation:** We evaluated GTraQEM's ability to generalize to larger circuits while training on the shallower circuits as detailed in Appendix 9.4.1.
>         - Trained on circuits with Trotter steps ranging from 1 to 15.
>         - Tested on circuits with Trotter steps from 16 to 20.
>         - Results presented in Fig 4c and the left column of Tab 2 demonstrate that GTraQEM maintains high performance on larger circuits, highlighting its ability to generalize across different circuit depths.
>
> We hope this clarifies the generalizability of GTraQEM.
>
> ***> "Q4: As this paper uses an IBM quantum computer (Brisbane), it is recommended to articulate the hardware's physical parameters to improve the soundness, e.g. the topology (including the 50 used qubits), the coherence time, the single-qubit and two-qubit fidelities during the experiment."***
>
> 1. Topology: IBM_Brisbane is a quantum computer with 127 qubits. We use the Qiskit quantum cloud platform, which first optimizes the 50-qubit circuit and maps it onto the 127 qubits using the ```generate_preset_pass_manager``` function. [This diagram](https://postimg.cc/wRN1T529) shows the topology structure of these 127 qubits.
> 2. Coherence time: Median T1 is 213.21 us, and median T2 is 147.61 us.
> 3. Fidelities: IBM_Brisbane provided the metric *2Q Error (layered)* to indicate the average two-qubit error rate per layered gate (EPLG) for a 100-qubit chain. 2Q Error (layered) of IBM_Brisbane is 2.06e-2.
> 4. Additional information: Median ECR error is 8.340e-3. Median SX error is 2.345e-4, Median readout error is 1.370e-2.
>
> We have added the detailed description of IBM_Brisbane to our updated pdf. Thanks for your suggestion.

---

> ### Author Response · Authors · 2024-11-20
> **Responses to Reviewer PHEy (3/3)**
>
> ### Responses to the Weaknesses
>
> ***> W1: Unclear experiment settings...***
>
> Thank you for bringing this to our attention. Due to space limitations, it was not possible to include all details here. Our mitigation efforts cover small-scale circuits (4/6 qubits), medium-scale circuits (15 qubits), and large-scale circuits (50 qubits) under four distinct error settings. A detailed introduction to our dataset preparation and noise model construction is included **in Appendix A 10.3** of our newly uploaded pdf.
>
> For questions related to reproducibility and ZNE performance, please check the response for Q1.
>
> ***> W2: The paper writing needs to be strengthened. For example, the 15-qubit circuits experiment in Table 2 seem to be confusing. Why do we need this?***
>
> The purpose of including 15-qubit circuits in Tab. 2 is to show that GTraQEM consistently maintains its mitigation performance as the number of qubits grows, showcasing its applicability and efficacy in larger quantum systems. We apologize for the ambiguity and have added a more detailed analysis of the results in the updated manuscript.
>
> ***> W3: Figure 3 is unclear. I can hardly recognize the color.***
>
> We already changed the color for clarity in our updated pdf.
>
> ***> W4: There are many typos. For example, 6 qubits should be 6-qubit (in Fig. 3), and the broken sentence in line 408.***
>
> We refined the writing in our updated pdf.
>
> ***> W5: The limitations of the proposed approach should be discussed, including the extra cost caused by the acquisition of the training set, and the cost of the training process.***
>
> We apologize for not previously including a discussion of the limitations of GTraQEM. The two costs you mentioned do indeed exist; however, they are inherent to all ML-based QEM strategies due to the shared mitigation paradigm, which differs from classical QEM methods. We included a discussion of the costs, along with other potential limitations of our work **in Appendix A13** of the newly uploaded pdf.
>
> We hope that our responses resolve your concerns, and if you have further concerns please let us know. We will be glad for further discussion.

---

> ### Author Response · Authors · 2024-11-24
>
> Dear Reviewer PHEy,
>
> Thank you for your valuable comments on our paper. Since the discussion phase deadline nears, we hope our responses have addressed your concerns. If needed, we are happy to provide more responses.
>
> Best wishes,
>
> the authors

---

> ### Comment · Reviewer_PHEy · 2024-11-24
>
> Thank you for your insightful reply. Your responses have addressed my concerns.
>
> I believe the revised manuscript and the open-source repository could significantly contribute to applying the deep learning method in quantum error mitigation. To help the development of quantum computing, deep learning methods are of great importance, and the recent advancements have clearly shown this trend, e.g. AlphaQubit by the Google team. Therefore, I recommend this paper be presented at ICLR.

---

> > ### Author Response · Authors · 2024-11-24
> >
> > Dear Reviewer PHEy,
> >
> > We sincerely appreciate your positive evaluation of our manuscript. We want to thank you again for the time and effort you dedicated to reviewing our manuscript and for the constructive feedback that helped improve its quality.
> >
> > Thank you once again for your support of our work.
> >
> > Best wishes,
> >
> > The authors

---

### Official Review · Reviewer_xXV6 · 2024-11-03

**Soundness:** 3
**Presentation:** 3
**Contribution:** 3
**Rating:** 6
**Confidence:** 4

**Summary:**

The paper proposes a novel approach, GTraQEM, for quantum error mitigation. This method is based on a graph transformer that employs multi-head attention mechanisms and a learnable Quantum Circuit-Representative node instead of traditional message-passing. The authors claim that this model captures both the circuit structure and its intrinsic nonlocality, which are limitations in existing learning-based or ML-based QEM methods. Experimental results demonstrate that GTraQEM outperforms ML-based approaches on both random quantum circuits and Trotterized circuits.

**Strengths:**

1. GTraQEM introduces a unique non-message-passing graph transformer approach and a circuit-to-graph encoding scheme for quantum error mitigation.
2. The authors present a data augmentation technique, called circuit inverse composition, which reduces dependency on ideal expectation values in the training data.
3. The paper demonstrates GTraQEM’s error mitigation performance compared to learning- or ML-based approaches on both random and structured quantum circuits across various noise types.

**Weaknesses:**

1. While the authors compare GTraQEM with learning- or ML-based QEM approaches, showing many promising results, classical QEM methods such as Probabilistic Error Cancellation, Symmetry Constraints, and Purity Constraints are not included in the experimental comparisons or discussed anywhere in the paper.
2. The authors clearly demonstrate that GTraQEM can achieve better performance in noise calibration compared to learning- or ML-based QEM methods. However, they do not compare the overheads of these methods, which is crucial for evaluating QEM methods.
3. I am concerned about the scalability of GTraQEM due to its reliance on edge features that require expectation measurements. As quantum circuits grow in size and complexity, the need for extensive measurements could result in significant resource demands, potentially limiting the applicability of GTraQEM for larger quantum systems or more complex circuits.
4. Typo: [Czarnik, 2021a] and [Czarnik, 2021b] refer to the same paper.

**Questions:**

My questions address the weaknesses of the paper:
1. Why did you choose not to include classical QEM methods such as Probabilistic Error Cancellation, Symmetry Constraints, and Purity Constraints in your experimental comparisons?
2. Can you provide experimental or theoretical comparison of the overhead associated with GTraQEM compared to other QEM methods?
3. How do you plan to address potential scalability issues related to the need for expectation measurements in GTraQEM as circuit sizes and complexities increase?
4. Could you explain why there is only ZNE compared in the no idea EV cases?

---

> ### Author Response · Authors · 2024-11-20
> **Responses to Reviewer xXV6 （1/3）**
>
> We sincerely appreciate the reviewer for the valuable comments. We set out below our responses to each of the questions.
>
> ### Responses to the Questions
> ***> Q1: "Why did you choose not to include classical QEM methods such as Probabilistic Error Cancellation, Symmetry Constraints, and Purity Constraints in your experimental comparisons?"***
>
> We previously decided not to include these three methods primarily due to their inherent disadvantages, such as high time complexity, qubit overheads, or limited generality. Namely,
>
> 1. **Probabilistic Error Cancellation (PEC)** [1] requires detailed characterization of the noise channels present in quantum hardware, which demands significant time and computational resources to acquire and maintain the noise matrices. This extensive characterization process can hinder scalability and broad applicability in larger quantum systems.
> 2.	**Purity Constraints** (mainly implemented via the method called **Virtual Distillation, VD** [2]) require the addition of circuit modules to monitor the original circuit, similar to approaches in quantum error correction. This accordingly causes considerable qubit overhead, specifically 2n + 1 qubits for an n-qubit circuit.
> 3.	**Symmetry Constraints** are typically utilized in Variational Quantum Eigensolver (VQE) problems, as they include constructing symmetry operators based on the system’s Hamiltonian [3]. In contrast, this manuscript focuses on achieving high generality across various quantum circuits.
>
> To address your concerns and further validate the efficacy of GTraQEM, we now include additional experimental results including PEC and VD as competitors from two experimental settings. The results are summarized below.
>
> #### Experimental Settings
>
> | Circ Type     | # of Circs (Train/Test) | Noise Types                         | Results                             |
> | ------------- | ------------------------- | ----------------------------------- | ----------------------------------- |
> | All-4-Trotter | 900 / 300                 | Coherent Noise                      | [here](https://postimg.cc/ZCHmjBwQ) |
> | All-6-Trotter | 900 / 300                 | Mixed Noise | [here](https://postimg.cc/23f3KWxj) |
>
> *Note that the y-axis in violin plots utilizes the symlog scale for clearer visualization.*
>
> #### Results
>
> | Method      | All-4-Trotter MAE | All-4-Trotter CIR | All-6-Trotter MAE | All-6-Trotter CIR |
> |-------------|-------------------|-------------------|-------------------|-------------------|
> | **GTraQEM** | **0.0619**        | 0.8533            | **0.0315**        | **0.9276**           |
> | ZNE         | 0.0974            | **0.8667**            | 0.2746            | 0.7900            |
> | CDR         | 0.0826            | 0.7933            | 0.0803            | 0.8967            |
> | ML-GNN      | 0.0699            | 0.8167            | 0.0397            | 0.8500            |
> | MLP         | 0.0779            | 0.8467            | 0.0581            | 0.8667            |
> | OLS         | 0.0813            | 0.7833            | 0.0491            | 0.8600            |
> | VD          | 0.0869            | 0.8600            | 0.0566            | 0.9167            |
> | PEC         | 0.1756            | 0.4467            | 0.2203            | 0.7633            |
> | Noisy       | 0.1324            | N/A               | 0.3227            | N/A               |
>
> The Mean Absolute Error (MAE) and Circuit Improvement Ratio (CIR) indicate that **GTraQEM** reduces the MAE compared to other methods, including the newly added baselines. While VD shows competitive performance, its significant qubit overhead makes it less practical for larger circuits (**incapable of performing mitigation for 15-qubit with appropriate execution time**). PEC, on the other hand, is sensitive to noise settings and requires extensive calibration efforts with unsatisfactory performance.
>
> We hope this clarifies our methodology and reinforces the efficacy of GTraQEM, and we will include the discussion in the updated manuscript.
>
> **References:**
>
> [1] *Probabilistic Error Cancellation with Sparse Pauli-Lindblad Models on Noisy Quantum Processors, Nature Physics, 2023.*
>
> [2] *Virtual Distillation for Quantum Error Mitigation, Physical Review X, 2021.*
>
> [3] *Low-Cost Error Mitigation by Symmetry Verification, Physical Review A, 2018.*

---

> ### Author Response · Authors · 2024-11-20
> **Responses to Reviewer xXV6 （2/3）**
>
> ***> Q2: Can you provide an experimental or theoretical comparison of the overhead associated with GTraQEM compared to other QEM methods?***
>
> We apologize for not including an overhead analysis in the original manuscript. The overhead analysis of QEM methods usually regards two aspects: **time complexity** and **qubit overhead**.
>
> #### 1. **Time complexity**
>
> We apologize that the manuscript did not include the time complexity analysis, as comparing the computational time between ML-based and classical QEM methods poses challenges due to **inherent differences in their workflows**.
>
> - Classical QEM approaches handle circuits **one at a time**, perhaps involving extensive mitigation time due to circuit-individual tuning, multiple circuit executions, or extra qubit overheads. Further, most of them are easily influenced by the number of qubits and the scale of circuits.
> - ML-based approaches involve two stages, **model training** and **inference**. **Once the model is well-trained, the inference (i.e., the actual mitigation) occurs extremely quickly.** For example, after fine-tuning GTraQEM for 6-qubit Trotter circuits executed under the Fake Hanoi Provider, we can mitigate the results of new 6-qubit Trotter circuits on that device in negligible time. Accordingly, a direct and fair comparison of these two types of methods is not straightforward.
>
> We now provide a table showing the training and inference times required by each ML-based method, alongside the mitigation times for ZNE, CDR, VD, and PEC applied to the same number of circuits during inference for comparison. **The table demonstrates that GTraQEM achieves efficiency comparable to other ML-based methods while showing stronger mitigation performance.** While GTraQEM’s inference is marginally slower than MLP and MLGNN, all three methods achieve inference times under one second, rendering the difference insignificant.
>
> |             | 4q - 300circs |               | 6q - 300circs |               | 15q - 100circs |               |
> | ----------- | ------------- | ------------- | ------------- | ------------- | -------------- | ------------- |
> | **Method**  | **Training**  | **Inference** | **Training**  | **Inference** | **Training**   | **Inference** |
> | **GTraQEM** | 21.90s        | 0.1760s       | 38.70s        | 0.1868s       | 18.64s         | 0.121s        |
> | **OLS**     | 3.42s         | 0.0734s       | 5.53s         | 0.0864s       | 2.21s          | 0.022s        |
> | **ML-GNN**  | 31.57s        | 0.1473s       | 43.54s        | 0.1916s       | 26.41s         | 0.104s        |
> | **MLP**     | 25.57s        | 0.0896s       | 32.71s        | 0.1037s       | 15.94s          | 0.098s        |
> | **ZNE**     | n/a           | 548.78s       | n/a           | 782.03s       | n/a            | 2166.46s     |
> | **CDR**     | n/a           | 1841.34s      | n/a           | 1796.65s      | n/a            | 17405.38s     |
>
> *Note: All quantum circuit simulations are executed via IBM qiskit. GTraQEM is implemented on a computing system running Ubuntu 18.04.6, equipped with CUDA version 12.1 The majority of experiments are performed on an NVIDIA GeForce RTX 4090 GPU, which has 24 GB of memory.*
> #### 2. **Qubit Overhead**
> - **GTraQEM and ML-Based Methods**: These methods do not require any additional qubits beyond those used in the original circuits.
> - **Classical QEM Methods**:
>   - **Virtual Distillation (VD)**: VD requires significant qubit overhead, scaling as $2n + 1$ qubits for an $n$-qubit circuit. This overhead can be impractical for larger circuits and exceeds the capacity of many current quantum devices.
>   - **Other Methods (ZNE, CDR, PEC)**: Generally, these methods do not require additional qubits but may involve increased circuit depth or additional circuit executions.
>
> In summary, despite the initial training phase, GTraQEM's **rapid inference time** results in lower total computational time compared to classical methods, especially as the number of circuits/qubits increases. GTraQEM also does not incur any qubit overhead, making it suitable for implementation on current quantum hardware without additional resource constraints. We included this detailed analysis in the revised manuscript.
>
>
> ***> Q3: “How do you plan to address potential scalability issues related to the need for expectation measurements in GTraQEM as circuit sizes and complexities increase?"***
>
> In our circuit, we only use the results of Pauli measurements on each qubit, which are considered **local observables**. By utilizing the classical shadow algorithm [4], we can reduce the original $M$ measurements to only $\log(M)$ measurements, and this reduction is independent of the number of qubits in the quantum circuit. In this way, we can address potential scalability issues as circuit sizes increase. We included this explanation in the revised manuscript.
>
> *[4] Predicting Many Properties of a Quantum System from Very Few Measurements, Nature Physics, 2020.*

---

> ### Author Response · Authors · 2024-11-20
> **Responses to Reviewer xXV6 （3/3)**
>
> ***> Q4: "Could you explain why there is only ZNE compared in the no idea EV cases?"***
>
> We apologize for any confusion regarding this setting. The "no ideal EV" scenario was initially designed to evaluate our circuit inverse composition technique, which aims to **reduce the reliance on ideal EVs as training labels** in ML-based QEM methods for the **large-qubit circuits**. We recognize that utilizing this technique on 6-qubit circuits can be somehow purposeless, as classical simulations are still feasible and efficient for such systems.
>
> The initial inclusion of the 6-qubit “no ideal EV” experiment aimed to offer a glimpse into the mitigation of large-qubit circuits，**where ZNE stands out as the only practical method available within a reasonable execution time for generally structured circuits.**
>
> Namely, our baseline methods can be categorized into classical and ML-based approaches:
>
> - **Classical Methods:** ZNE is one of the few QEM techniques that can be applied to large-qubit systems within a reasonable time frame. Other methods face significant challenges as the number of qubits increases. For instance, VD involves substantial qubit overhead, while PEC and CDR are typically time-consuming, e.g., when we employ PEC provided by Qiskit to mitigate 50-qubit circuit's noise on the IBM real quantum devices, the submitted task will time out. These factors hinder their applicability in large-scale quantum systems.
>
> - **ML-Based Methods:** Although ML-based QEM methods are time-efficient and do not require additional qubits, they traditionally rely on ideal EVs of training circuits as training labels. For instance, in small-scale settings, we generate training circuits similar in structure to the testing circuits and perform classical simulations to obtain these labels for supervised learning. However, as the number of qubits increases, obtaining ideal EVs becomes impractical. Our study accordingly addresses this challenge by utilizing the circuit inverse composition technique to reduce dependence on ideal EVs, thereby making ML-based QEM feasible for large-qubit systems.
>
> To better align with these considerations, we have decided to remove the 6-qubit "no ideal EV" experiment from the main body of our paper. Instead, we will **include a detailed analysis of this setting as above, along with the experiment of "6-qubit no ideal EV" and all relevant baseline methods as standard settings, in the appendix**. This adjustment allows us to focus on demonstrating the effectiveness of GTraQEM under the refined learning paradigm. Notably, we have performed error mitigation on 50-qubit circuits executed on IBM Brisbane, which showcases our method with this refined paradigm efficacy in large-scale quantum systems in Fig. 4 (f) in the manuscript, and we present additional results of ZNE for this setting in [this figure](https://postimg.cc/wybH0g9D)
>
> Thank you again for your valuable feedback. Your comments helped us improve the clarity and focus of our manuscript, and we will update the revised version in a couple of days.
>
> ### Responses to the Weaknesses
>
> ***> W1: "While the authors compare GTraQEM with learning- or ML-based QEM approaches, showing many promising results, classical QEM methods such as ... in the paper."***
>
> Please look at our answer to Q1.
>
> ***> W2: "The authors demonstrate ... compare the overheads of these methods, which is crucial for evaluating QEM methods."***
>
> Please look at our answer to Q2.
>
> ***> W3: "I am concerned about the scalability of GTraQEM ..."***
>
> Please look at our answer to Q3.
>
>
> ***> W4: Typo: [Czarnik, 2021a] and [Czarnik, 2021b] refer to the same paper.***
>
> Thank you for pointing this out! We already fixed it in the manuscript.
>
>
> We hope our responses effectively address your concerns. If you have any questions or need further clarification, please let us know, and we will gladly discuss further.

---

> > ### Comment · Reviewer_xXV6 · 2024-11-23
> >
> > Thank you for the insightful reply. Can you comment on the comparison of sampling overhead for your methods comparing to others, it is okay just for inference stage. Thank you!

---

> ### Author Response · Authors · 2024-11-24
> **Discussion about the sampling overhead**
>
> Thank you for your reply and insightful question. We would like to comment on the comparison of sampling overhead for our method.
>
> 1. **Sampling Overhead Consideration**
>
> **We consider that sampling overhead is not a suitable metric for quantifying the cost of ML-based QEM methods, as generally no additional circuit runs (i.e., shots) are required for mitigation in the context of these methods.** To elaborate on this, we first introduce the explicit definition of sampling overhead. Specifically, the goal of error mitigation is to construct an estimator $\overline{O_{\text{em}}}$ such that it achieves a **smaller bias** than the naive noisy estimator $\overline{O_{\rho}}$, i.e.,
> $$
> |\text{Bias}[\overline{O_{\text{em}}}]| \leq |\text{Bias}[\overline{O_{\rho}}]|.
> $$
> Yet, reducing the bias often leads to an increase in the variance of the estimator:
> $$
> \text{Var}[\overline{O_{\text{em}}}] \geq \text{Var}[\overline{O_{\rho}}].
> $$
>
> As the number of circuit runs needed for a given estimator $\hat{X}$ to achieve a shot noise level $\epsilon$ is given by:
> $$
> N_{\text{shot}}^{\epsilon}(\hat{X}) = \frac{\text{Var}[\hat{X}]}{\epsilon^2},
> $$
>
> for traditional QEM methods, to achieve $\epsilon$-precision with the mitigation estimator, the **sampling overhead** is defined as:
> $$
> C_{em} = \frac{N^\epsilon_{\text{shot}}(\hat{O_{\text{em}}})}{N^\epsilon_{\text{shot}}(\hat{O_{\rho}})}.
> $$
>
> *Note that we follow all the definitions provided in [1]; please refer to Section II B in [1] for detailed explanations of each notation.*
>
> However, because no additional circuit executions ($N^\epsilon_{\text{shot}}(\hat{O_{\text{em}}})$ ) are required during the mitigation phases, the concept of sampling overhead is ambiguous for ML-based methods like GTraQEM. Namely, in GTraQEM, circuit executions are performed during the data preparation stage, where noisy EVs of both training and testing circuits are collected as features, and the ideal EVs of the training circuits serve as labels. **Consequently, the number of shots needed for mitigation is identical to that of the raw noisy execution, making it unclear how to define** $N^\epsilon_{\text{shot}}(\hat{O_{\text{em}}})$.
>
>
> We also found that [2], which proposes multiple ML-based QEM methods, does not quantify the sampling overhead either. Instead, **they focus on circuit runtime overhead, quantifying the number of circuits required to perform mitigation.**
>
> 2. **Runtime Overhead Analysis**
>
> Following the approach in [2], we present the runtime overhead of GTraQEM. Specifically, for mitigation over $N_{\text{test}}$ circuits during the inference stage, the runtime overhead is compared among methods as shown in the table below:
>
> | Method                         | Circuit Runtime Overhead             |
> | ------------------------------ | ------------------------------------ |
> | **ZNE**                           | up to ~ $M \times N$                 |
> | **PEC**                            | up to ~ $2 \times N$                 |
> | **ML-based (GTraQEM/MLP/GNN/OLS)** | $N_{\text{train}} + N_{\text{test}}$ |
>
> The values for ZNE and PEC come from Table IV in [1]. Since $N_{\text{train}}$ is set and only needs to be performed once for circuits with similar structures under one noise setting, ML-based methods enjoy efficiency with smaller runtime overhead compared to traditional QEM techniques, especially when the test datasets enlarges, as demonstrated in [2].
>
> **References**
>
> [1] Quantum Error Mitigation, Rev. Mod. Phys. 95, 045005.
>
> [2] Machine Learning for Practical Quantum Error Mitigation, Nat Mach Intell (2024).
>
> ---
>
> We hope this reply addresses your concerns. If you have other questions, we would love to have a further discussion.

---

> ### Comment · Reviewer_xXV6 · 2024-11-24
>
> Thank you for addressing my concerns. I would like to update my rating to a higher score.

---

> > ### Author Response · Authors · 2024-11-25
> >
> > Dear Reviewer xXV6,
> >
> > We sincerely appreciate your decision to increase the rating of our manuscript! Your valuable feedback and insightful suggestions help refine our work. Thank you once again for your time and consideration.
> >
> > Best regards,
> >
> > The Authors

---

### Official Review · Reviewer_6ert · 2024-11-04

**Soundness:** 3
**Presentation:** 3
**Contribution:** 2
**Rating:** 6
**Confidence:** 4

**Summary:**

The authors put forward a new graph-transformer method for quantum error mitigation. By not using message passing, the approach has the potential to better capture the long-range correlations created by quantum circuits.

**Strengths:**

Strengths:
   - It is a nice application of graph transformers. While it’s pretty straightforward, their approach works better than other GNN-based quantum error mitigation methods that I’ve seen.
   - Pretty good results on simulated data. While a lot of the error bars overlap, the new method regularly outperforms competitors on many of the toy error models.

**Weaknesses:**

REVISION: Many of my concerns have been addressed.

Weaknesses:
   - Lots of misleading or slightly incorrect claims in the background section. See below.
   - Simulations leave a fair bit to be desired. In particular, they don’t use any error models with coherent errors, which are particularly pernicious kinds of errors because they’re effects depend intimately on the unitary implemented by a quantum circuit (e.g., coherent errors can cancel). The IBM fake providers use an error model that models gate errors as a depolarizing channel followed by thermal relaxation (T1 decoherence). Their incoherence-only error model only features depolarizing noise, which is probably the easiest to mitigate
   - No head-to-head comparisons on experimental data: They ran 50-qubit circuits on IBM Brisbane but then didn’t compare their new approach to other methods (if they did, I’m sorry! I missed it in the paper).

Miscellaneous remarks:
   - The circuit-to-DAG encoding (or a very similar encoding) used in this paper was used earlier in this paper: “QuEst: Graph Transformer for Quantum Circuit Reliability Estimation.” You should cite it.

Here’s a list of the misleading claims in the paper:
   - “Quantum Error Correction (QEC) (Calderbank & Shor, 1996; Gottesman, 1997; Terhal, 2015) first offers a theoretical solution by fully correcting quantum errors at the hardware level, but its implementation demands impractically qubit overheads and complex operations (Cai et al., 2023).” I don’t think that QEC corrects errors at the hardware level. Most QEC protocols (e.g., quantum error correcting codes) require syndrome data to be extracted and then processed by a classical co-processor. The quantum state is then adaptively updated (at least when running a circuit that uses a universal gate set) in response to the results of the syndrome data analysis. I’d change this statement to better reflect how QEC works.
     I also don’t think it’s fair to say that QEC is “impractical.” Large-scale fault-tolerant quantum computation is certainly currently infeasible, but lots of teams are working on making it a reality! I’d re-phrase this to say that large-scale fault-tolerance quantum computation is currently well beyond the capabilities of our experimental hardware.
   - “ Overall, QEM provides a feasible approach to enable imperfect quantum systems to produce reliable outcomes (Kandala et al., 2019; Bravyi et al., 2022; Cai et al., 2023), which is crucial for achieving practical quantum supremacy over classical supercomputers (Daley et al., 2022; Kim et al., 2023).” So, most quantum error mitigation methods also have an exponential-in-the-qubit-count overhead. So it’s not clear if they’ll offer a path to quantum computational supremacy. Past works, such as in the Kim et. Al. Paper, have only shown hints at quantum advantage, which is a much weaker claim. This sentence should be rewritten to reflect this.
   - “ Machine learning-based QEM methods have recently been developed, offering greater generality across various settings.” This claim is questionable and requires citations.
   - “ Bell nonlocality (Bell, 1964; Brunner et al., 2014) demonstrates that entangled particles exhibit correlations where the measurement outcome of one particle instantaneously influences that of another, regardless of the distance between them.” This is not an accurate description of Bell nonlocality. Entanglement does not allow particles to influence each other at a distance. No information is transmitted. Instead Bell nonlocality allows for measurement distributions with non-classical correlations.
   - “ We introduce a data augmentation technique that constructs training data by composing circuits with their inverse circuits.” Composing circuits with their inverses in order to (ideally) create the identity circuit is not a new idea. It underpins almost all randomized benchmarking algorithms (Clifford RB, Magann et. Al.). You should at least mention this.
   - Section 2.2 gives a very non-standard overview of the noise sources in quantum computers. For instance, I’ve never heard anyone use the term “real quantum device errors” before. I would re-write the section to first talk about Markovian vs. non-Markovian errors, then step through the differences between the kinds of Markovian errors a device can experience: incoherent/stochastic errors, coherent errors, and “other errors” like amplitude damping (see “A Taxonomy of Small Markovian Errors”). N.B. I’m pretty sure that all quantum error channels also have a Kraus decomposition (this is not clear from your claim in A.7.1 when you state that incoherent errors have a Kraus decomposition).

**Questions:**

Why are there no comparisons to other techniques in the 50-qubit data?

What was the total wall clock time spent running each error mitigation method in your simulations?

---

> ### Author Response · Authors · 2024-11-20
> **Response to Reviewer 6ert (1/3)**
>
> Thank you for your review and acknowledgment of our work. Here are our specific responses to refine our paper based on your comments.
>
> ### Responses to the Questions
>
> ***> Q1: Why are there no comparisons to other techniques in the 50-qubit data?***
>
> We apologize for the oversight. We have provided additional comparative 50-qubit experimental results for **MLGNN**, **MLP**, and **ZNE**, as shown in [this figure](https://postimg.cc/wybH0g9D) and the following table. The other baseline, OLS, requires more resources than are available on our current devices, so we did not include it in the comparison. For the ZNE method, we used the ```Estimator_v2``` integrated into Qiskit, which can be used to mitigate the noise of quantum circuits executed on IBM real quantum hardware. We did not compare with CDR because it is very time-consuming, and we do not have enough free usage of the IBM quantum platform for the entire mitigation. From the table, we can see that GTraQEM still outperforms other baselines for the 50-qubit circuits on IBM Brisbane.
>
> | Method      | MAE (Mean Absolute Error) | CIR (Circuit Improved Ratio) |
> | ----------- | ------------------------- | ---------------------------- |
> | **GTraQEM** | **0.3943**                | **0.83**                     |
> | ML-GNN      | 0.5798                    | 0.67                         |
> | MLP         | 1.0309                    | 0.11                         |
> | ZNE         | 0.7125                    | 0.45                         |
> | Noisy       | 0.7342                    | N/A                          |
>
>
> ***> Q2: What was the total wall clock time spent running each error mitigation method in your simulations?***
>
>
> We apologize that the manuscript did not include the time complexity analysis, as comparing the computational time between ML-based and classical QEM methods poses challenges due to **inherent differences in their workflows**.
>
> - Classical QEM approaches handle circuits one at a time, perhaps involving extensive mitigation time due to circuit-individual tuning, multiple circuit executions, or extra qubit overheads. Further, most of them are easily influenced by the number of qubits and the scale of circuits.
> - ML-based approaches involve two stages, **model training** and **inference**. **Once the model is well-trained, the inference (i.e., the actual mitigation) occurs extremely quickly.** For example, after fine-tuning GTraQEM for 6-qubit Trotter circuits executed under the Fake Hanoi Provider, we can mitigate the results of new 6-qubit Trotter circuits on that device in negligible time. Accordingly, a direct and fair comparison of these two types of methods is not straightforward.
>
> We now provide a table showing the training and inference times required by each ML-based method, alongside the mitigation times for ZNE and CDR applied to the same number of circuits during inference for comparison. **The table demonstrates that GTraQEM achieves efficiency comparable to other ML-based methods while showing stronger mitigation performance.** While GTraQEM’s inference is marginally slower than MLP and MLGNN, all three methods achieve inference times within one second, rendering the difference insignificant.
>
>
> |             | 4q - 300circs |               | 6q - 300circs |               | 15q - 100circs |               |
> | ----------- | ------------- | ------------- | ------------- | ------------- | -------------- | ------------- |
> | **Method**  | **Training**  | **Inference** | **Training**  | **Inference** | **Training**   | **Inference** |
> | **GTraQEM** | 21.90s        | 0.1760s       | 38.70s        | 0.1868s       | 18.64s         | 0.121s        |
> | **OLS**     | 3.42s         | 0.0734s       | 5.53s         | 0.0864s       | 2.21s          | 0.022s        |
> | **ML-GNN**  | 31.57s        | 0.1473s       | 43.54s        | 0.1916s       | 26.41s         | 0.104s        |
> | **MLP**     | 25.57s        | 0.0896s       | 32.71s        | 0.1037s       | 15.94s          | 0.098s        |
> | **ZNE**     | n/a           | 548.78s       | n/a           | 782.03s       | n/a            | 2166.46s     |
> | **CDR**     | n/a           | 1841.34s      | n/a           | 1796.65s      | n/a            | 17405.38s     |
>
> *Note: All quantum simulations are executed on IBM Qiskit. GTraQEM is implemented on a computing system running Ubuntu 18.04.6, equipped with CUDA version 12.1 The majority of experiments are performed on an NVIDIA GeForce RTX 4090 GPU, which has 24 GB of memory.*
>
> We hope that the provided information elucidates our proposed method's computational efficiency compared to other error mitigation techniques. We will also add a detailed time complexity analysis in the updated manuscript.

---

> ### Author Response · Authors · 2024-11-20
> **Response to Reviewer 6ert (2/3)**
>
> ### Responses to the Weaknesses
>
> ***> Simulations leave a fair bit to be desired. In particular, they don’t use any error models with coherent errors, ... Their incoherence-only error model only features depolarizing noise, which is probably the easiest to mitigate***
>
> In our initial experiments, we considered three noise scenarios: incoherent noise alone, incoherent noise combined with readout errors, and IBM’s fake providers. We chose these models as they effectively capture key aspects of real quantum device noise—especially the last scenario—and provide meaningful insights for error mitigation.
>
> Moreover, we conducted experiments on a real quantum device, IBM’s Brisbane (please refer to our response to Q1), and the results continue to validate the effectiveness of our method. The simulation results were included to complement these real-device experiments by allowing for the analysis of a larger dataset and the exploration of diverse scenarios. Both real and simulated error settings demonstrated the practical applicability and robustness of our approach.
>
> To enhance evaluation robustness and address your concern, **we now include an additional experiment involving coherent errors.** Specifically, following the prior studies [1], we applied coherent over-rotations of 0.02$\pi$ radians to the CX, CY, CZ, and SWAP gates, combined with the existing noise model extracted from the fake Lima device. For this new experiment, we utilized the All-4-Trotter circuit set, which includes 900 circuits for training and 300 for testing. The distribution of AE is displayed in [this figure](https://postimg.cc/fkMgCH1w), and the MAE and CIR for each method are detailed below:
>
> | Method   | MAE        | CIR        |
> | -------- | ---------- | ---------- |
> | **Ours** | **0.0619** | 0.8533 |
> | ZNE      | 0.0974     | **0.8667**     |
> | CDR      | 0.0826     | 0.7933     |
> | ML-GNN   | 0.0699     | 0.8167     |
> | MLP      | 0.0779     | 0.8467     |
> | OLS      | 0.0813     | 0.7833     |
> | Noisy    | 0.1324     | N/A        |
>
>
> The extra experiment demonstrates that our approach remains effective while including the coherent errors, as it shows the best MAE and second best CIR among baseline methods, proving its robustness over the mitigation of coherent error.
> We hope this resolves your concerns and shall include the results in our manuscript to deliver a more comprehensive evaluation.
>
> **References:**
> [1] Machine Learning for Practical Quantum Error Mitigation,  Nat Mach Intell (2024).
>
> ***> No head-to-head comparisons on experimental data: They ran 50-qubit ...***
>
> Please check our reply for Q1.
>
>
> ***> The circuit-to-DAG encoding (or a very similar encoding) used in this paper was used earlier in this paper: “QuEst: Graph Transformer for Quantum Circuit Reliability Estimation.” You should cite it.***
>
> We apologize for the unclear illustration of our encoding scheme. While both GTraQEM and Quest map each quantum gate to a graph node, this is a standard approach in graph learning for quantum computing, as demonstrated in prior works like [3] and [4]. We have built upon these studies to develop our unique scheme. We clarify the distinctions between our encoding scheme and that of [2]:
>
> 1. **Virtual Quantum Circuit-Representative (QCR) Node**: Our encoding introduces a **virtual QCR node that is adjacent to all physical nodes in the graph**, which not only alters the structure but also fundamentally impacts the model's aggregation scheme, setting our approach apart from [2]. The efficacy of including a QCR node rather than a standard pooling layer is demonstrated in the ablation studies presented in Appendix A.13 in the manuscript.
>
> 2. **Edge Features with Qubit-wise Measurement Outcomes**: We incorporate qubit-wise measurement outcomes directly into the graph as edge features, providing additional information that enhances the model's performance. In contrast, the encoding in [2] does not utilize edge features.
>
> 3. **Distinct Node Features**: The node features in the GTraQEM model differ significantly from those in [2].
>
> We will update the manuscript to highlight our unique encoding scheme and incorporate appropriate citations to related works, including [2], [3], and [4]. This will help readers grasp the novelty of our encoding scheme and its contributions to the field
>
> **References:**
>
> [2] QuEst: Graph Transformer for Quantum Circuit Reliability Estimation, ICCAD 2022.
>
> [3] Graph Neural Network Autoencoders for Efficient Quantum Circuit Optimisation, arXiv 2023.
>
> [4] A GNN-based Predictor for Quantum Architecture Search, Quantum Information Processing, 2023.

---

> ### Author Response · Authors · 2024-11-20
> **Response to Reviewer 6ert (3/3)**
>
> ***> “ We introduce a data augmentation technique that constructs training data by composing circuits with their inverse circuits.” Composing circuits with their inverses to (ideally) create the identity circuit is not a new idea...***
>
> We apologize for the unclear statement in our manuscript. While creating identity circuits by composing circuits with their inverses is a well-established technique in other quantum computing fields, **our contribution is, to our best knowledge, the first to use this method to reduce dependence on ideal EVs in training ML-based QEM methods.**
>
> ML-based QEM has the model training and inference stages, and **the labels (ideal EVs) of the training circuits are required for supervision** to guarantee the model's performance during the inference (i.e., the mitigation) stage. However, obtaining ideal EVs as labels can be difficult, particularly when the qubit number of circuits is large and ML models usually require more data for training. Accordingly, such a lack of training labels limited the application of ML-based QEM models.
>
> In this manuscript, by employing circuit inverse composition, we enable **label-efficient training**, allowing the acquisition of ideal EVs of training circuits without execution, which greatly reduces the difficulty in obtaining labels as the scale of the circuit increases.
>
> In summary, this approach extends the applicability of ML-based QEM to larger qubit systems while maintaining performance via label-efficient training, as demonstrated with our 50-qubit experiments on IBM Brisbane. We view this as a novel contribution to our manuscript instead of first proposing this technique. We hope this clarifies your concern and will revise the manuscript to clearly state the point.
>
> **We already rephrased the sentence in our introduction.**
>
> ***> Here’s a list of the misleading claims in the paper...***
>
> Thank you for your helpful recommendations regarding our introduction and background. We updated the manuscript with revisions in highlights based on your suggestions.
>
> ---
> We hope our replies adequately address your concerns. Please let us know, and we would be happy to discuss further.

---

> ### Author Response · Authors · 2024-11-24
>
> Dear Reviewer 6ert,
>
> We sincerely appreciate your valuable feedback on our submission. As the discussion phase deadline nears, we look forward to your responses. If you have any additional questions or concerns, please let us know—we are happy to discuss them further.
>
> Thank you once again for your time and insights.
>
>
> Best wishes,
>
> the authors

---

> > ### Comment · Reviewer_6ert · 2024-11-25
> >
> > Looks good. Thanks for taking the time to address everyone's concerns.

---

> ### Author Response · Authors · 2024-11-25
>
> Dear Reviewer 6ert,
>
> We sincerely appreciate your support of our work and your willingness to raise your evaluation score. Your detailed suggestions, particularly those concerning the background information, have been invaluable in helping us refine our manuscript.
>
> Thank you once again for your thoughtful review.
>
> Best regards,
>
> The Authors

---

### Meta-Review · Area_Chair_29o4 · 2024-12-20

**Metareview:**

This paper introduces GTraQEM, a novel QEM method that leverages a graph transformer model with a non-message-passing architecture. The approach features a unique circuit-to-graph encoding scheme that integrates quantum-specific positional encoding, attention bias, and a learnable Quantum Circuit-Representative node. These innovations aim to effectively capture long-range dependencies and mitigate errors in quantum circuits. Experimental evaluations demonstrate that GTraQEM outperforms most standard QEM methods on both random and structured circuits, across diverse noise types and scales.

Reviewers highlighted the innovative integration of graph-transformer techniques with quantum-specific enhancements for QEM. They also acknowledged the strong empirical results, particularly the method's effectiveness for larger-scale circuits. Overall, GTraQEM presents interesting insights and a promising direction for the quantum computing community.

**Additional Comments On Reviewer Discussion:**

During the rebuttal period, the authors and reviewers engaged in discussions addressing key concerns, including the reproducibility of results due to the absence of publicly available code, the limited experimental comparisons with classical QEM methods, and the scalability of GTraQEM to larger quantum circuits. Reviewers also questioned the novelty of certain proposed techniques and sought clarification on the method's limitations, particularly regarding training dataset requirements and computational overhead.

In response, the authors made substantial efforts to address these concerns. They released an anonymous repository containing code snippets and data to enhance reproducibility. Additional experimental results were provided, incorporating classical QEM methods as baselines. The authors clarified the novelty of their circuit-to-graph encoding scheme and scalability issues. Furthermore, they revised the manuscript to improve clarity, added detailed discussions of the method’s limitations, and included updated experimental results with more robust comparisons. While some concerns about practical hardware implementation remain, the authors and reviewers reached a consensus on most critical issues.

---

### Decision · Program_Chairs · 2025-01-22

Accept (Poster)